# Zooarchaeological Evidence from Medieval Ojców Castle, Lesser Poland

**Joanna Religa-Sobczyk** [1], **Krzysztof Wertz** [1], **Lembi Lõugas** [2], **Michał Wojenka** [3], **Anna Lemanik** [1] **and Piotr Wojtal** [1,*]

1 Institute of Systematics and Evolution of Animals of the Polish Academy of Sciences, Sławkowska 17, 31-016 Krakow, Poland

2 Archaeological Research Collection, Tallinn University, Rüütli 10, 10130 Tallinn, Estonia

3 Institute of Archeology, Jagiellonian University in Kraków, Gołębia 11, 31-007 Krakow, Poland

* Correspondence: wojtal@isez.pan.krakow.pl

**Abstract:** Archaeological research at Ojców castle has yielded important information about life in that medieval castle. The results of zooarchaeological analyses presented in this paper complement the archaeological research, adding to our knowledge of the diet of the castle inhabitants from the time of establishment of the castle until the final residents. Zooarchaeological research is also complemented by data from older settlement phases on the castle hill, directly related to the Lusatian culture in the early Iron Age. The great variability of remains from mammals, birds, and fish and the taphonomic features of bones found in the different chronological strata of the castle's courtyard reflect the diverse economic activities that took place in particular times and spaces.

**Keywords:** zooarchaeology; archaeology; taphonomy; castle; husbandry; Iron Age; Lusatian culture; Middle Ages; Poland





## 1. Introduction

The castle under study is located in Ojców, Skała commune (Lesser Poland), on a limestone hill, in the karstic-rich landscape of Prądnik Valley, in the so-called Polish Jura (Figure 1). Archaeological excavations and research on the castle Ojców were initiated because it is one of the most impressive defensive fortifications of the Polish Middle Ages and because of its poor state of repair and the scarcity of written accounts related to its genesis [1]. The oldest traces of settlement on the castle hill discovered so far are those of the late Lusatian culture [2], predating the castle. Subsequent stages of the settlement are directly related to the medieval castle. Written sources indicate that the castle's founder was Polish king Casimir III the Great [2], and the activities in the castle began most probably at the second half of the 14th century. According to a legend, the fortified castle in the Prądnik Valley was named by Casimir the Great "Ociec u Skały" (*Ociec*/*Ojciec* in Polish means father) in honor of Władysław I Łokietek (father of Casimir III the Great) wandering in the local forests and caves [3]. Hence the likely current name of the castle in Ojców.

After the death of Casimir the Great (1370), the castle Ojców and its surroundings often changed owners as tenure, and, up until the early 19th century, it was held by noblemen, albeit still formally being a royal property. In the 16th century, the castle was temporarily in the hands of Andrzej Tęczyński, Queen Bona, and Stanisław Płaza of Mstyczów. The loss of the defensive role of the castle and lack of funds for maintenance led, in 1619, to the sale of Ojców district to a royal secretary Mikołaj from Pilica Koryciński. Over the next few years, the castle, under the care of the Koryciński family, Topór coat of arms, took on a new splendor. The peak of its status coincides with the rules of Stefan Koryciński, a great royal chancellor since 1652 [3]. Even the Swedish deluge did not lead to the destruction of the castle, because, as sources indicate, at that time it became a warehouse for weapons and food for the invaders [3]. Years later, successive generations of the Koryciński family took

greater care of the grounds of Łęczyca, where they settled permanently. With the consent of King John III, the castle was sold to Jan Kazimierz and Domicela of the Warszycki family, and later on it regularly changed hands. In 1703, the castle was sold again to the Stanisław and Helena Morski family. During the next 20 years, the Great Northern War swept through Poland and did not spare the castle. In the meantime, Helena Morska's brother Bogusław Łubieński and his wife became the owners of the castle.

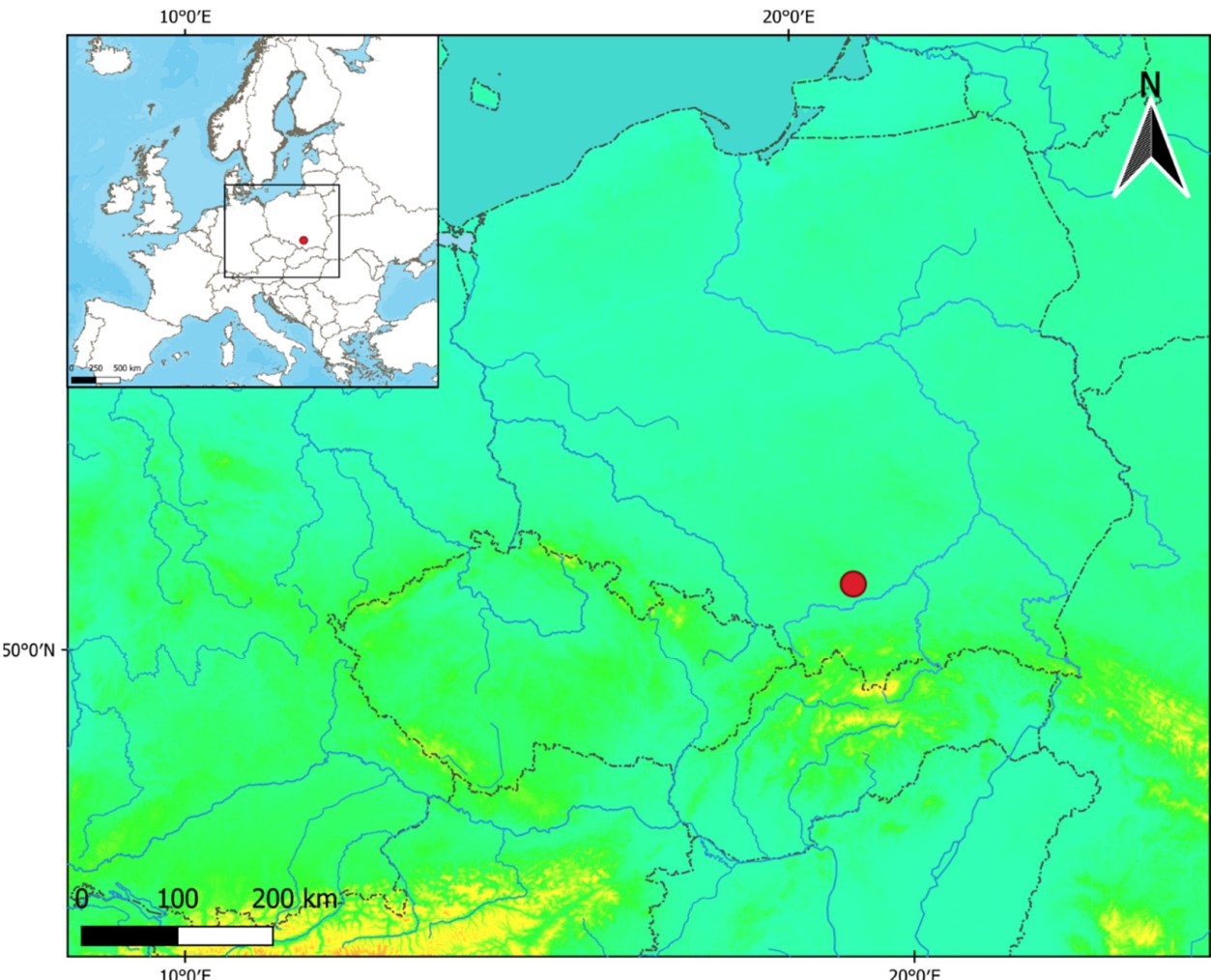

**Figure 1.** Map with location of the castle in Ojców.

Unfortunately, lack of current repairs led to ruin of what was rebuilt by Koryciński. Łubieński undertook another reconstruction project for his son Zygmunt and his wife Marianna Dembińska. After her husband's death, Marianna Dembińska married again to Ignacy Załuski. From the middle of the 18th century, the castle was home to the Załuski family. The piling up of financial problems eventually led to the loss of the castle. The end of the residence/functioning of the castle is attributed to the beginning of the 19th century.

Beyond the interesting history of changing ownership, the castle management needs attention because this building is remarkable in the landscape. Therefore, in this paper, the main aim is to give a comprehensive overview of the faunal remains found in the different parts of the castle. In addition, the comprehensive zooarchaeological analysis in this area is still rare. So far, the only known studies are mainly related to medieval Cracow [4,5]. Faunal remains commonly show the food preferences of the human occupants. Usually, a local origin of products is expected. However, in the case of the unique site Ojców, imports and animals kept as pets expectably might be present in the material. To meet the aim of this study, the animal remains are here sorted according to the phases that correspond to certain

periods when the castle was in use. The taxonomic as well as anatomical identifications are presented, as well as the taphonomy of the bones, i.e., the different traces and marks of the activities that have affected the animals and their remains.

## 2. Materials and Methods

The first archaeological excavations in the Ojców Castle took place in 1991 under the supervision of Krystyna Kruczek. More extensive archaeological investigations began in 2006, and they have been continued ever since by Michał Wojenka of Jagiellonian University. These excavations made it possible to recover the animal remains that are the subjects of this study, among other significant data. Since 2006, ten trenches have been explored in different parts of the castle courtyard (Figure 2).

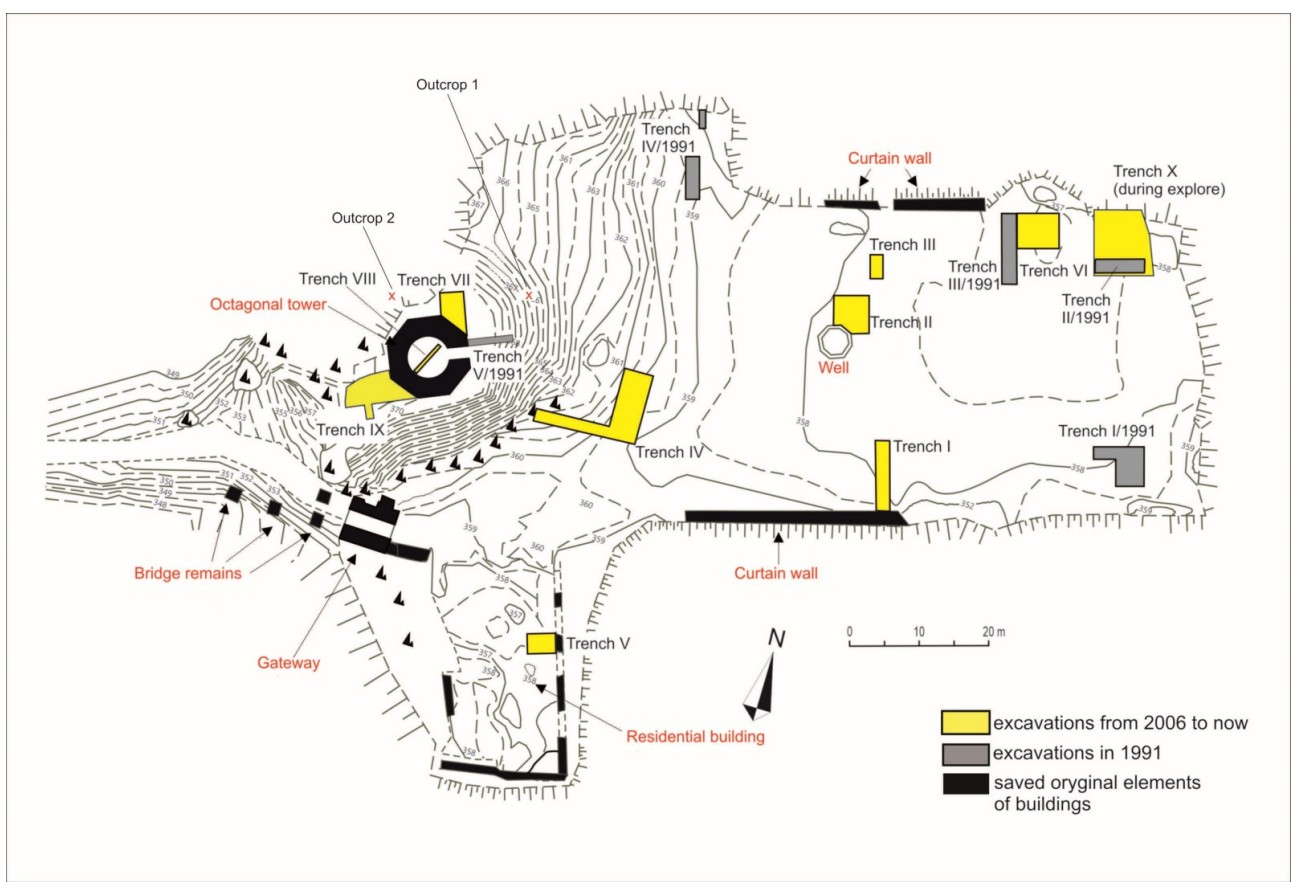

**Figure 2.** Map with the location of excavations within the castle walls (figure by Michał Wojenka).

*Chronology of the Sediments*

All trenches from where our animal samples came were located within the castle walls. The stratigraphy correlates in each trench. Detailed descriptions of layers and division by phases was already made for trench I and II [1], and it is valid for other trenches mentioned in this paper as well.

Phase I is related to the Early Iron Age Lusatian culture, dated from 900–750 BCE to the second half (or the end) of the 7th century BCE or the first half of the 6th century BCE.

Phase II marks the beginning of the castle constructions, including the erection of curtain walls, dated from the second half of the 14th century C.E. (=A.D.) to the turn of the 14th/15th century/beginning of the 15th century C.E.

Phase III relates to the medieval settlement with the curtain walls and part of the buildings, dated to the 15th century C.E.

Phase IV marks the post-medieval times with some renovation, which links to the Koryciński family and possibly earlier, in chronological periods from the 16th century to the first half of the 17th century C.E.

Phase V covers the late post-medieval period, from completion of the restoration works by the Koryciński family until the end of the function as a castle, all dated approximately from the mid 17th century to the very early 19th century C.E.

There are also mixed phases recorded during the excavations. Some of the osteological materials were collected from heterogeneous layers. Because of that, they cannot be related to any specific phase.

The osteological material was recorded in trenches I–IX. Some additional material came from the outcrops. During the excavations, bones of larger mammals and birds were collected manually; smaller animals such as rodents, fish, amphibians, and mollusks came from wet-sieved soil samples collected from each trench. The zooarchaeological studies of osseous material from trenches I and II were published earlier [1]. In this study, we present the complete results of all analyses, including the previously published ones. The assemblage of all animal bone material from excavations of Ojców Castle is stored in the Institute of Systematics and Evolution of Animals, Polish Academy of Sciences in Cracow.

The osteological material in this study includes remains of mammals, birds, and fish. Preservation of the osseous finds ranges from average to poor. Many fragments of animal remains were collected that cannot be identified to the lower taxonomic levels (genus and/or species). All of the determinable bones of mammals, birds, and fish were subjected to detailed zooarchaeological analysis. Remains were first taxonomically described on the basis of comparative material stored in the osteological collections of the Institute of Systematics and Evolution of Animals, PAS, anatomical atlases [6–14] and comparative skeletal collections available on the internet. Fish bones were identified in Tallinn University using the skeletal reference collection in the Archaeological Research Collection. Based on the cortical thickness of the mammal bone fragments and their dimensions, three categories of unidentifiable remains were possible to establish: large sized mammals (e.g., horse and cattle), medium sized mammals (e.g., goat/sheep, pig, dog), and small sized mammals (e.g., hare and fox). In the case of fish bones, approximate size categories were assessed, i.e., small, medium, and large mean sizes of fish within single taxa (not among fish in general). Due to the extensive fragmentation of the osteological material and the near lack of complete specimens, bone measurements were not made.

The birds' maturity was established based on stages of epiphyseal fusion and diaphysis porosity [15]. Bird bones were sexed based on traits such as the size of particular species or the presence of specific structures such as a tarsometatarsal spur or medullary bone [15].

The next step in the analysis was to detect the taphonomic processes that affected the bones. For that, the surfaces of bones were carefully inspected using a strong directional light in order to find possible post-mortem modifications. Traces of human activity connected with the processing of mammal carcasses were recorded, especially cut marks and chopping marks. Marks on the bones were verified under a microscope of high magnification and correlated with specific locations expected for different human actions (skinning, meat-stripping (=filleting), disarticulating) as described by Binford [16]. Chopping marks are often created during dismemberment, and they are easier to notice because, unlike cut marks, the chopping action often results in the fracture or separation of a piece of bone. Besides dividing the carcass into smaller parts, chopping was also aimed at accessing the marrow inside bones, which also resulted in bone breakage [17–20]. Another type of human activity recognizable on the remains is the use of fire for cooking, although sometimes burning marks can be a result of accidental contact with fire. Depending on the color of a burnt bone surface, the scale and time of fire action can be distinguished [21].

Other types of traces recognizable on bones are modifications made by carnivores. Because the collected material came from a human settlement site (Iron Age and Lusatian culture settlement; medieval and post-medieval castle), we are confident that domestic

dogs were responsible for the carnivore modifications on the mammal remains, specifically gnawing marks and partial digestion of bones.

Other modifications on mammal remains from Ojców Castle are trampling marks caused by human and/or animal activity [21–23]. Trampling marks were also verified under a microscope.

During the studies of the mammal osseous material, some biotic causes that affected the bones were recognized: e.g., the stages of modification caused by weathering, described here based on the criteria presented by A. Behrensmeyer [22]. In addition, traces were noted of physical and chemical processes such as calcite precipitations and root etching. Behrensmeyer [22] supposed that the latter is caused by "dissolution [of bone surfaces] by acids associated with the growth and decay of roots or fungus." Lyman [21] pointed out that it is unclear whether the acids that etch bone surfaces come from plant roots or from fungi on decomposing plant parts.

Bird bones were examined under a low-power microscope for traces of animal activity (gnawing, digesting, coprolite coating), human activity (e.g., tool usage, fire, marks of bone bending), and factors influencing already deposited bones (e.g., root etching, weathering, rodent gnawing, trampling) (see e.g., [15]). Tool-made marks made during "dismembering" include chop marks, cut marks on articular surfaces, and deep, short, transverse cuts just below epiphyses suggesting the severing of tendons. The remaining tool-made marks were harder to interpret and, therefore, classified simply as "cut marks." Gnawing traces made by rodents were clearly distinguishable. Gnawing traces by other animals could not always be distinguished among possible agents affecting smaller bones (namely dogs, cats, and humans).

Based on the numerical data, standard calculations in zooarchaeology were made for mammal remains. These include the Number of Identified Specimens (NISP) and the minimum number of individuals (MNI) [21,24,25]. Due to the variability in approaches to determining MNI, the method of separating individuals based on opposing parts of the skeleton was chosen without joining the individual parts of each given bone together.

Obtained MNI values of bird taxa respect sides of the body (left–right) and recorded bone zones [26] but not the maturity of the specimens. MNI values for fish remains were not calculated because of the dominance of vertebrae in the material, which would give an unrealistic result, because most vertebrae cannot be associated with specific individuals.

Due to the anatomical similarity between skeletons of the domestic goat (*Capra hircus*) and domestic sheep (*Ovis aries*), many caprine/ovine bones and teeth could not be assigned to the appropriate species. Such remains were placed in an inclusive goat/sheep category (*Capra hircus/Ovis aries*) [27–29].

## 3. Results

The studied osteological material (Table 1) consists of 15,580 bones and teeth of mammals, more than 625 bones of birds from at least 60 individuals and 18 taxa (Table 2), and 349 fish bones and more than 900 scales (Table 3). Only 23% of mammal remains were assigned to either domestic or wild species. The most numerous remains come from cattle, pig, and goat/sheep (Table 1, Figures 3 and 4). Not much—only 1% of the entire identifiable mammal remains—comes from wild animals. Within the wild animal subassemblage, hares are the most numerous. Unfortunately, the vast majority of all bones in the whole assemblage, ca. 77% (NISP = 11,952), were undeterminable remains, and could only be assigned to groups divided into size categories (Table 1).

**Table 1.** Distribution of mammal remains by NISP and MNI of individual species.

| Taxon | Chronology by Phases | | | | | | | | | | | Total Phases | | Total NISP |
|---|---|---|---|---|---|---|---|---|---|---|---|---|---|---|
| | Phase I | | Phase II | | Phase III | | Phase IV | | Phase V | | Mixed * | | | |
| | NISP | MNI | NISP | MNI | NISP | MNI | NISP | MNI | NISP | MNI | NISP | NISP | MNI | |
| Beaver (*Castor fiber*) | | | | | 1 | 1 | | | | | 1 | **1** | **1** | 2 |
| Rabbit (*Oryctolagus cuniculus*) | | | | | | | | | 1 | 1 | 1 | **1** | **1** | 2 |
| Hare (*Lepus europaeus*) | 1 | 1 | | | 7 | 1 | 17 | 2 | 20 | 2 | 23 | **45** | **6** | 68 |
| Felidae | | | | | 1 | 1 | | | | | | **1** | **1** | 1 |
| Red fox (*Vulpes vulpes*) | | | | | 3 | 1 | | | | | | **3** | **1** | 3 |
| Wolf (*Canis lupus*) | | | | | 1 | 1 | | | | | 5 | **1** | **1** | 6 |
| Dog (*Canis familiaris*) | 9 | 2 | | | 26 | 2 | 3 | 1 | 2 | 1 | 12 | **40** | **6** | 52 |
| Horse (*Equus caballus*) | 15 | 1 | | | 16 | 1 | 1 | 1 | 8 | 1 | 10 | **40** | **4** | 50 |
| Roe deer (*Capreolus capreolus*) | 2 | 1 | | | 5 | 1 | | | 2 | 1 | 4 | **9** | **3** | 13 |
| Red deer (*Cervus elaphus*) | 3 | 1 | | | 7 | 1 | 1 | 1 | 4 | 1 | 3 | **15** | **4** | 18 |
| Elk (*Alces alces*) | | | | | | | | | | | 1 | | | 1 |
| Goat/sheep (*Capra hircus/Ovis aries*) | 61 | 2 | | | 155 | 3 | 151 | 4 | 137 | 3 | 217 | **504** | **12** | 721 |
| Cattle (*Bos taurus*) | 132 | 4 | 8 | 1 | 375 | 11 | 188 | 10 | 258 | 8 | 482 | **961** | **34** | 1443 |
| Pig (*Sus domesticus*) | 75 | 4 | 6 | 1 | 380 | 8 | 204 | 9 | 219 | 4 | 354 | **884** | **26** | 1238 |
| Wild boar (*Sus scrofa*) | 2 | 1 | | | 1 | 1 | 1 | 1 | 1 | 1 | 4 | **5** | **4** | 9 |
| Bear (*Ursus arctos*) | 1 | 1 | | | | | | | | | | **1** | **1** | 1 |
| **Identifiable total** | **301** | | **14** | | **978** | | **566** | | **652** | | **1117** | **2511** | | **3628** |
| Small sized mammals | 15 | | 2 | | 17 | | 64 | | 76 | | 99 | **174** | | 273 |
| Medium sized mammals | 209 | | 9 | | 467 | | 377 | | 612 | | 686 | **1674** | | 2360 |
| Large sized mammals | 342 | | 15 | | 804 | | 395 | | 839 | | 1084 | **2395** | | 3479 |
| Unidentifiable | 623 | | 12 | | 1522 | | 1023 | | 1238 | | 1422 | **4418** | | 5840 |
| **Unidentifiable total** | **1189** | | **38** | | **2810** | | **1859** | | **2765** | | **3291** | **8661** | | **11,952** |
| **Total NISP/MNI** | **1490** | **18** | **52** | **2** | **3788** | **33** | **2425** | **29** | **3417** | **23** | **4408** | **11,172** | **105** | **15,580 ** |

* MNI for mixed phases was not counted, ** NISP without rodents.

**Table 2.** Distribution of bird remains by NISP and MNI of individual species.

| Taxon | Chronology by Phases | | | | | | | | | | | Total | |
|---|---|---|---|---|---|---|---|---|---|---|---|---|---|
| | Phase I | | Phase II | | Phase III | | Phase IV | | Phase V | | Mixed Phases | | |
| | NISP | MNI | NISP | MNI | NISP | MNI | NISP | MNI | NISP | MNI | NISP | NISP | MNI |
| Goose (*Anser* sp.) | 6 | 1 | 2 | 1 | 7 | 1 | 7 | 1 | 61 | 5 | 32 | **115** | 9 |
| cf. *Anser* sp. | | | | | | | | | 1 | | 2 | **3** | |
| Mallard (*Anas platyrhynchos*) | | | | | | | 1 | 1 | 3 | 1 | 7 | **11** | 21 |
| cf. *Anas platyrhynchos* | | | | | 2 | 1 | | | 2 | | 3 | **7** | |
| Anseriformes (duck/goose) | | | | | | | | | 1 | | | **1** | |
| Peafowl (*Pavo cristatus*) | | | | | | | | | 1 | 1 | 3 | **4** | 1 |
| cf. Galliformes (small size) | | | | | | | | | 1 | 1 | | **1** | 1 |
| Domestic chicken (*Gallus domesticus*) | 35 | 5 | 3 | 1 | 21 | 4 | 33 | 5 | 103 | 11 | 109 | **304** | 26 |
| cf. *Gallus domesticus* | 1 | | | | 2 | | 5 | | 10 | | 19 | **37** | |
| Hazel grouse (*Tetrastes bonasia*) | 1 | 1 | | | | | | | | | | **1** | 1 |
| Capercaillie (*Tetrao urogallus*) | | | | | 1 | 1 | | | | | | **1** | 1 |

**Table 2.** *Cont.*

| Taxon | Chronology by Phases | | | | | | | | | | | Total | |
|---|---|---|---|---|---|---|---|---|---|---|---|---|---|
| | Phase I | | Phase II | | Phase III | | Phase IV | | Phase V | | Mixed Phases | | |
| Galliformes (middle size) | | | | | 1 | | 6 | | 28 | +3 | 9 | **44** | **3** |
| Rock pigeon (*Columba livia*) | 1 | 1 | | | | | | | | | 1 | **2** | **1** |
| *Columba livia/oenas* | | | | | 1 | 1 | | | | | | **1** | **1** |
| a Pigeon (*Columba* sp.) | | | | | | | | | 2 | 1 | | **2** | **1** |
| Eurasian woodcock (*Scolopax rusticola*) | | | | | 1 | 1 | | | | | | **1** | **1** |
| Sparrowhawk (*Accipiter nisus*) | 1 | 1 | | | | | | | | | 1 | **2** | **1 (+1)** |
| Goshawk (*Accipiter gentilis*) | | | | | | | | | | | 1 | **1** | **(+1)** |
| Common kestrel (*Falco tinnunculus*) | | | | | | | | | 2 | 1 | 1 | **3** | **1** |
| Falcon (*Falco* sp.) | | | | | 1 | 1 | | | 1 | | | **2** | **1** |
| Eurasian jay (*Garrulus glandarius*) | 1 | 1 | | | | | | | | | | **1** | **1** |
| Magpie (*Pica pica*) | | | | | | | | | 1 | 1 | | **1** | **1** |
| Jackdaw (*Corvus monedula*) | | | | | | | | | 3 | 1 | | **3** | **1** |
| Corvid (*Corvidae* indet.) | 1 | | | | | | | | | | | **1** | |
| House sparrow (*Passer domesticus*) | | | | | | | 1 | 1 | | | 1 | **2** | **1** |
| Woodlark (*Lullula arborea*) | | | | | | | | | 1 | 1 | | **1** | **1** |
| Song thrush (*Turdus philomelos*) | 1 | 1 | | | | | | | | | | **1** | **1** |
| Unidentified bird (*Aves* indet.) | 5 | | 1 | | 8 | | 9 | | 16 | | 15 | **54** | |
| Aves indet. (big size) | | | | | | | | | 3 | | 2 | **5** | |
| cf. Aves indet. | | | | | 2 | | | | 2 | | 9 | **13** | |
| **Total NISP/MNI** | **53** | **11** | **6** | **2** | **47** | **10** | **62** | **8** | **242** | **27** | **215** | **625** | **60** |

**Table 3.** Fish bones and scales (sq) divided between different phases (NISP is presented).

| Taxon | Chronology by Phases | | | | | | Total |
|---|---|---|---|---|---|---|---|
| | Phase I | Phase II | Phase III | Phase IV | Phase V | Mixed Phases | |
| Sturgeon (*Acipencer* sp.) | | 3 | | | | 1 | **4** |
| Eel (*Anguilla anguilla*) | | | | 1 | | | **1** |
| Herring (*Clupea harengus*) | | | 22 | | | 2 | **24** |
| Pike (*Esox lucius*) | 2 | | 21 | 3 + 1 sq | 5 | 8 | **39 + 1 sq** |
| Perch (*Perca fluviatilis*) | | | 31 | | | 2 | **33** |
| Percidae | | | 430 sq | 15 sq | | 27 sq | **472 sq** |
| Salmonidae | | | 2 | | | | **2** |
| Carp (*Cyprinus carpio*) | | | | | | 1 | **1** |
| Bream (*Abramis brama*) | | | 2 | | | | **2** |
| Ide (*Leuciscus idus*) | | | 1 | | | | **1** |
| Dace (*Leuciscus leuciscus*) | | | | 1 | | | **1** |
| Roach (*Rutilus rutilus*) | | | 2 | 1 | | | **3** |
| Tench (*Tinca tinca*) | | | | 1 | | | **1** |
| Cyprinidae | 1 | | 36 + 360 sq | 3 | 3 | 16 + 16 sq | **58 + 376 sq** |

**Table 3.** *Cont.*

| Taxon | Chronology by Phases | | | | | | Total |
|---|---|---|---|---|---|---|---|
| | **Phase I** | **Phase II** | **Phase III** | **Phase IV** | **Phase V** | **Mixed Phases** | |
| Burbot (*Lota lota*) | 1 | | 4 | | | | **5** |
| Pisces indet. | | | 124 + 45 sq | 11 + 15 sq | 1 | 28 + 27 sq | **164 + 87 sq** |
| **Total:** | **4** | **3** | **245 + 835 sq** | **21 + 31 sq** | **9** | **66 + 70 sq** | **349 + 936 sq** |

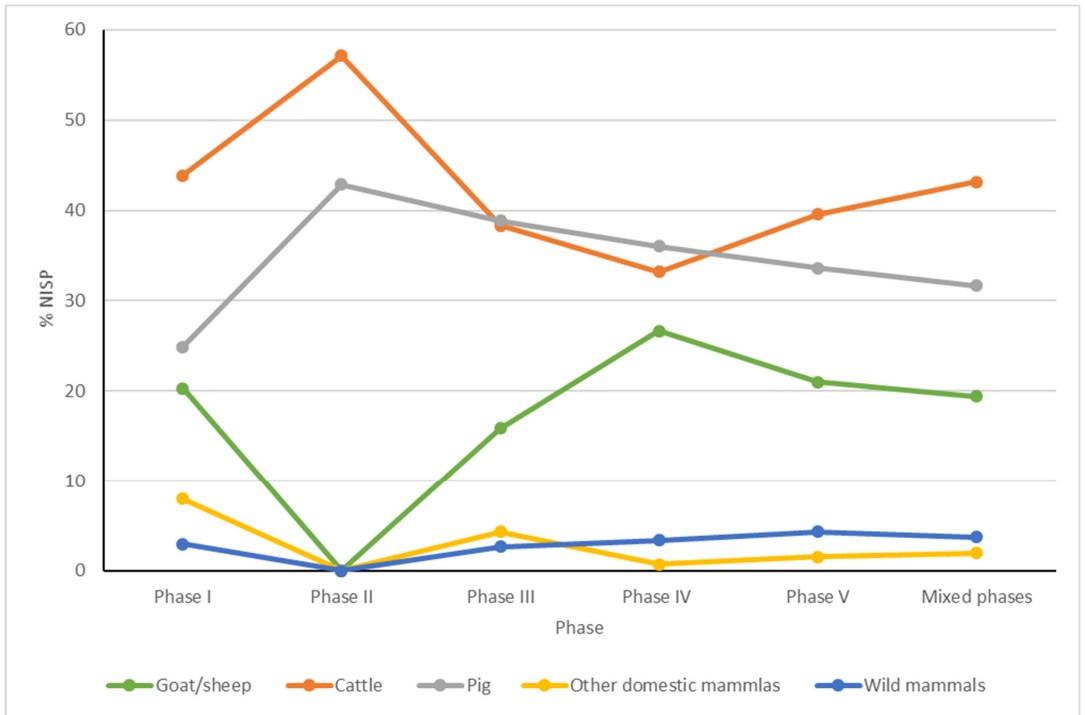

**Figure 3.** Percentage of Number of Identified Specimens (NISP) selected groups of mammals (other domestic mammals—horse, dog) discovered in main phases of settlement of the Ojców Castle.

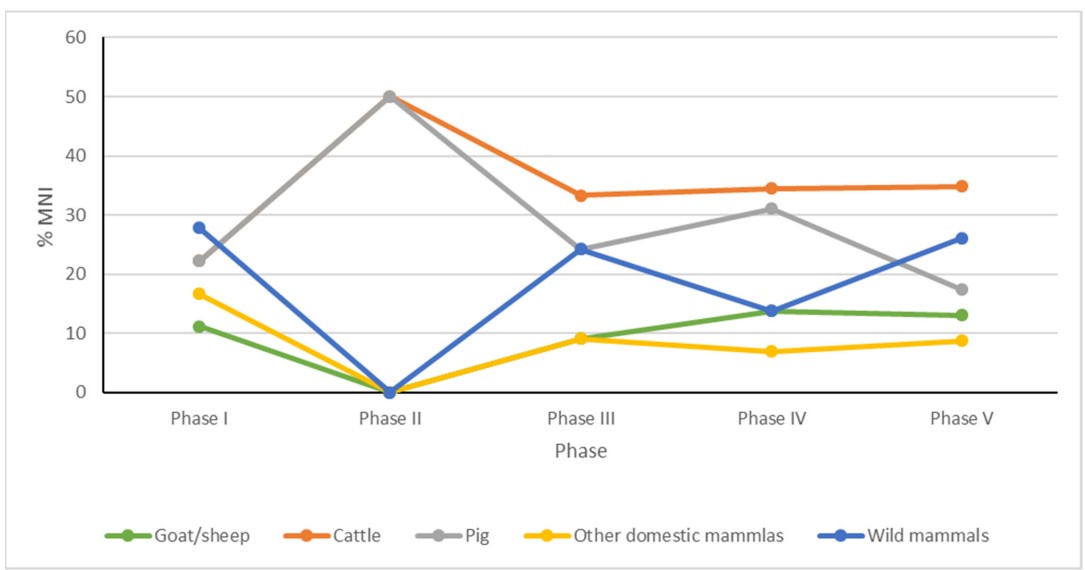

**Figure 4.** Percentage of minimum number of individuals (MNI) selected groups of mammals (other domestic mammals—horse, dog) discovered in main phases of settlement of the Ojców Castle.

Two taxa dominate in the bird bone assemblage: the domestic chicken (almost 50% of total bird NISP) and goose (almost 20% of total bird NISP) (Figures 5 and 6).

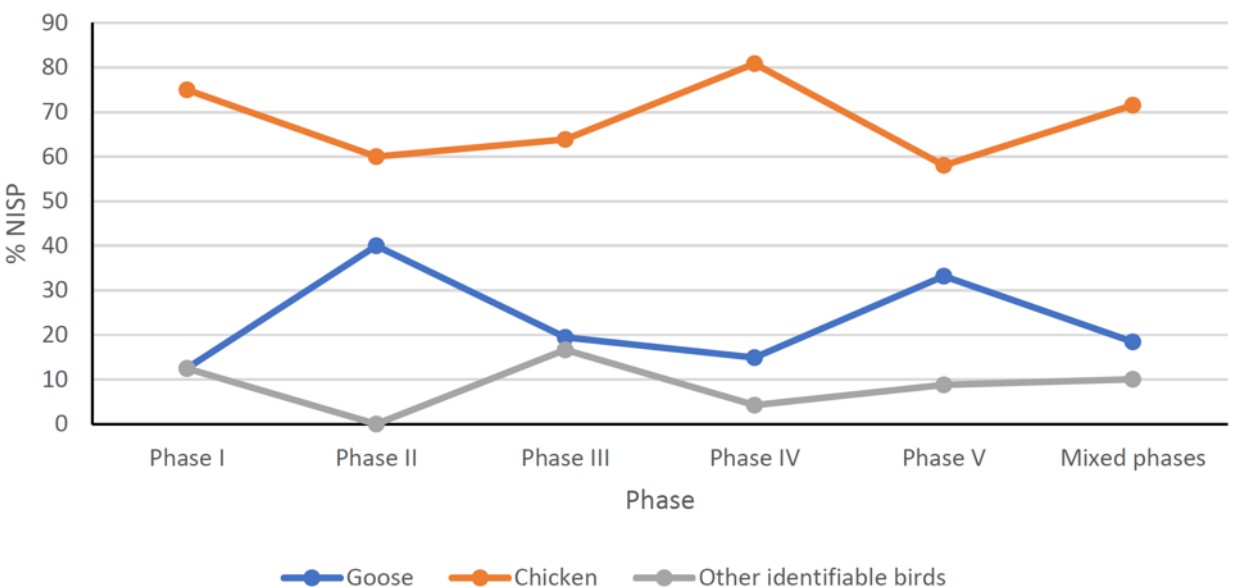

**Figure 5.** Percentage of Number Identified Specimens (NISP) selected groups of birds discovered in main phases of settlement of the Ojców Castle.

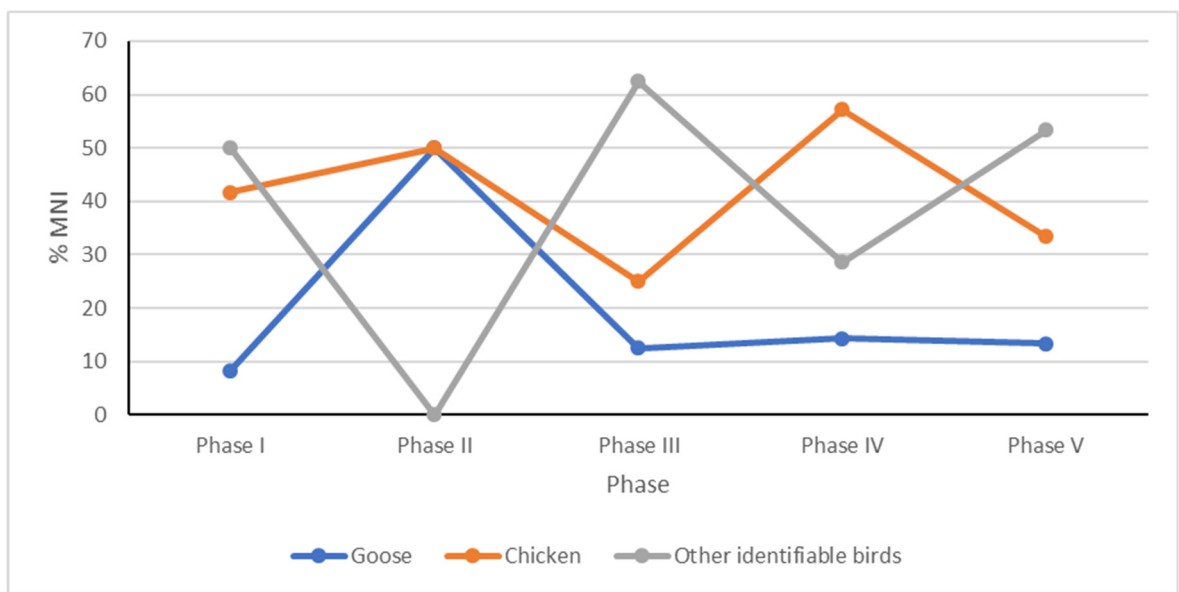

**Figure 6.** Percentage of minimum number of individuals (MNI) selected groups of birds discovered in main phases of settlement of the Ojców Castle.

Fish were represented by freshwater fish such as burbot (*Lota lota*), perch (*Perca fluviatilis*), and pike (*Esox lucius*). Very numerous is a group of cyprinid fish. Few bones of the cyprinids were identified to the species: the carp (*Cyprinus carpio*), dace (*Leuciscus leuciscus*), ide (*Leuciscus idus*), tench (*Tinca tinca*), roach (*Rutilus rutilus*), and common bream (*Abramis brama*). In addition, some marine and migratory fish were present such as herring (*Clupea harengus*) (most probably the Baltic herring), eel (*Anguilla anguilla*), sturgeon (*Acispenser* sp., cf. *A. oxyrinchus*), and representatives of family *Salmonidae*. There were also many fish bones (mainly fragments of ribs and fin rays) not assigned to any species (see Table 3).

### 3.1. Phase I

In the Phase 1 sediments, which were registered in trenches II, IV, VI, and VII (Figure 2), 1490 teeth and bones of mammals were discovered (Table 1). About 20% could be identified to genus or species. The greatest number of remains from domestic mammals is from cattle. The other well represented taxa (pig and goat/sheep) have roughly equal proportions in NISP. Only single specimens of horse and dog were found and also only single remains of three wild mammal taxa (red deer, roe deer, wild boar) (Table 1).

Signs of human activity (cut marks, chopping marks, burned bones) were observed on ca. 10% of mammal bones (Table A1). Cut marks (n = 62) related to skinning, dismembering, and filleting carcasses are recorded only on the bones of cattle, pig, and goat/sheep (Figure 7a; Table A1). On the remains without precise identification, cut marks were noted on 48 specimens. Among the signs of human activity on mammal remains from this phase, the most abundantly observed marks were those of chopping (n = 70). One mark was on the goat/sheep horn core, and another on antlers of the red deer. A chop mark was also on the horse pelvis and the dog metacarpal, which was interpreted as made during skinning [1]. The number of chopping traces on the remains not assigned to a precise species (n = 44) was similar to the number of cut marks recorded in the same phase. Traces of fire (burning marks) on the bones are also related to human activity. These were observed only on the bones of domestic mammals (n = 5) and on the bones without precise identification (Figure 7b) (n = 36) (Table A1).

Besides the signs that clearly indicate human actions on the mammal bones from this phase, the gnawing traces of carnivores (undoubtedly dogs) and rodents also were observed (Table A2). The gnawing marks made by dogs (n = 138) (on ca. 9% of bones in this phase) and the traces of digestion (n = 7) were recorded mostly on the remains of domestic herbivore species (Table A2). However, gnawing marks were discovered on a fragment of a dog scapula too. It should be noted that also one femur has signs left by dog teeth. Traces of digestion were only on the limb bones of the goat/sheep. The rest of the bones with digestion marks were fragments not assigned to any precise species.

We would like to emphasize that the largest number of bones with rodent gnawing marks were observed in this phase (n = 21). Marks made by rodent teeth were identified on a fragment of cattle mandible and pelvic bone, shafts of long bones from goat/sheep, and a distal part of a pig humerus. Signs of rodent activity were visible also on fragments of long bones, vertebrae, and pelvic bones of mammals without precise identification.

There were only a few bones with trampling marks (n = 5) (Table A2), root etching (n = 7), and signs of weathering (n = 5) discovered from Phase I (Table A3).

Fifty-three bird bones of at least seven taxa came from this phase (Table 2), the most numerous being the chicken. Fourteen chicken bones belonged to females, another one to a male, while six bones belonged to immature specimens. The sparrowhawk's bone came from a female. Two bone fragments of the goose may belong to the same element (right radius); both contain medullary bone structure, meaning the bird(s) died during an egg-laying period, probably between March and May. The bones of the Eurasian jay and the song thrush might have been deposited without human assistance. Both birds are synanthropic but were also kept alive by people during the Middle Ages for their singing [30]. The presence of the rock pigeon in this phase is perplexing and was discussed by Wojtal et al. [1]. A very high relative number of bird bones (almost 40%) bear tool marks, indicating the dismembering of bird carcasses (Table A4). A single chicken bone was gnawed by a rodent (Tables A5 and A6), but no other bones have a clear trace of animal activity (such as gnawing or digesting).

Only four fish bones were found. One represents a medium-to-large size cyprinid, two represent pike, and one is from burbot (Table 3).

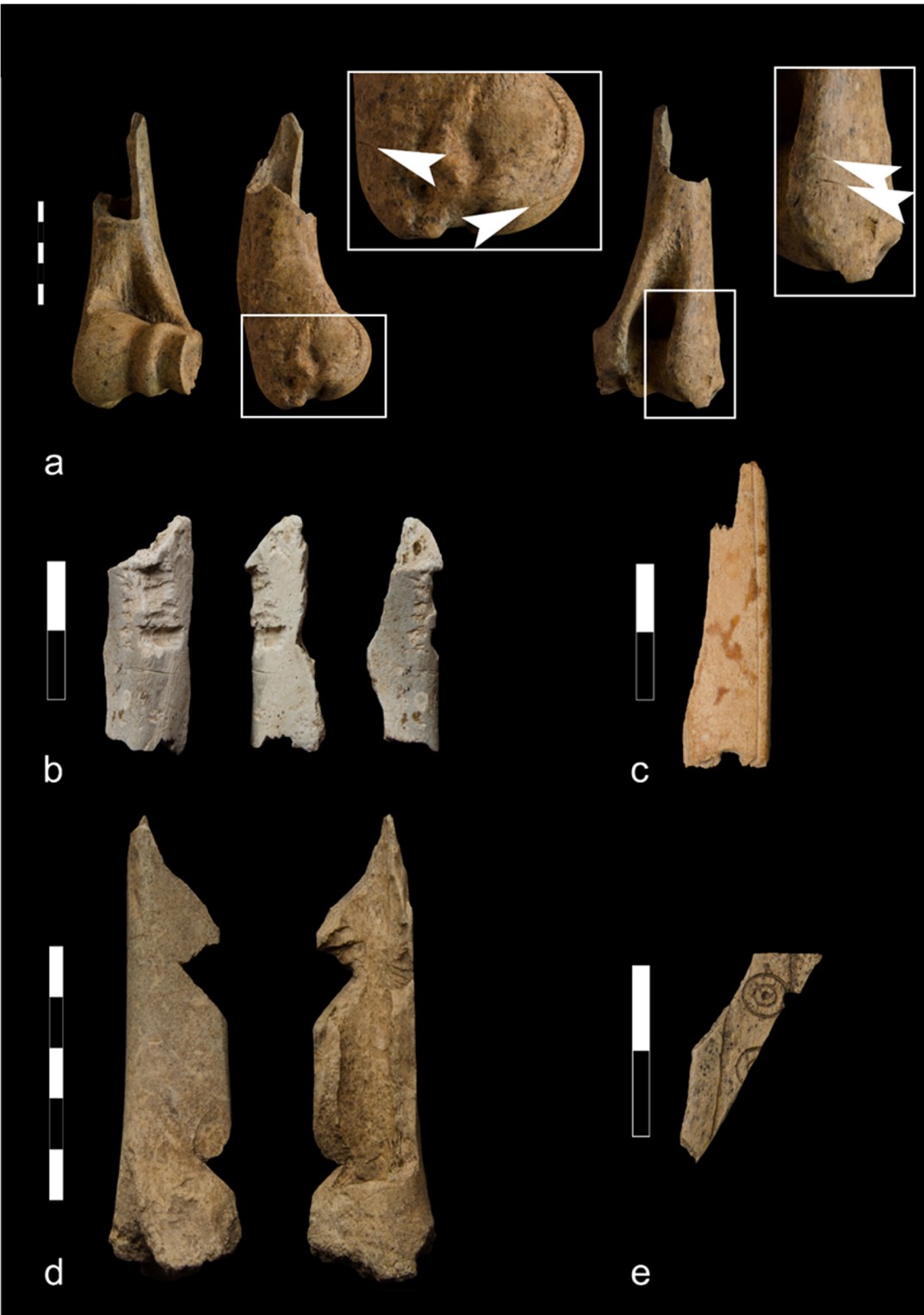

**Figure 7.** Bone finds from Ojców castle: (**a**) Goat/sheep humerus with cut marks on distal part associated with carcass dismembering (phase I); (**b**) Long bone of the medium size mammal burnt to white/grey color with cut mark and rodent biting on the shaft (phase I); (**c**) Fragment of bone object (probably bone handle) (phase III); (**d**) A pig tibia with lateral chopping marks across the shaft (phase III); (**e**) Fragment of bone object with ornament (phase IV).

### 3.2. Phase II

The Phase II sediments yielded the smallest amount of mammal remains (n = 52). Bones were discovered only in trenches I and VI. Cattle and pig represent the taxa identified to species. (Table 1, Figures 3 and 4).

The signs of human activity are infrequent and chopping marks are the most numerous (n = 8) (Table A1). There are also two bones with cut marks and only one burned pig bone fragment.

Signs of carnivore activity are also not numerous (n = 10). Trampling marks were observed on only one vertebra not assigned to a species (Table A2).

Despite the small amount of osteological material collected from this phase, there are proportionately many (n = 6; 11.5%) with calcite precipitations (Table A3).

Only six bird bones were discovered in sediments of Phase II. Besides one unidentified bone, they all belong to chicken and goose; the bones of the latter are distal elements of the wing (carpometacarpus and phalanx *digiti majoris*). On the goose carpometacarpal bone, there is a clear chop mark accompanied by an evident strong cut mark, possibly being a surface mark made before the final chop. At least one of the chicken bones belonged to a hen, and another one to an immature specimen.

Interestingly, despite of the small collection of fish remains, a dental bone and two scutae (broken into five pieces) of rather large Atlantic sturgeon (*Acipenser oxyrinchus*) (NISP = 3) are associate with this phase [1]. Unfortunately, it is not possible to tell whether all the discovered bones came from the same individual or more than one fish is represented.

### 3.3. Phase III

Mammalian osteological materials from Phase III were collected in trenches I, IV, VI, and IX; the highest number of mammalian remains (NISP = 3788) were recorded from the layers of this phase and also the largest number of taxa (n = 13) among all the phases. The largest number of domestic mammal bones came from cattle (MNI = 11), as in every phase. It is also noted that the largest number of wild animal taxa was found in Phase III. The remains of beaver, hare, fox, roe deer, red deer, and wild boar were identified (Table 1). The most frequent traces of human activity are chop marks (n = 273), followed by cut marks (n = 52); the least frequent traces are burned bones (n = 24) (Table A1). The chopping marks (Figure 7d) present more frequently than cut marks on all the remains of domestic animals (Table A1). We note that almost half of these marks are on pig bones (n = 12), especially on skull bones (the maxilla and the symphysis of the mandible). Chopping marks were also observed on three of the red deer remains. Traces of fire are visible only on 24 mammal bones; among them is a burnt fragment of a dog mandible.

Phase III contains the largest number of bones gnawed by dogs (n = 851; 22% of total) and bones digested (by dogs) (n = 169, ca. 4%). Most of the gnawing marks were found on the bones of cattle, pig, and goat/sheep, but some were also seen on the long bones of horse and dog. Despite the low number of remains of wild mammals, the marks of dog teeth were observed on long bones of roe deer, red deer, and wild boar. It should be also mentioned that there were two digested long bones of hare. The rodent gnawing marks are very rare, and they are visible only on three bones: on a pig calcaneus and on two bones not assignable to any certain species. The trampling marks, which are usually visible on the bone surface, are not common here and were found only on 24 fragments (Table A2). Different stages of weathering were observed mostly on cattle bones. Other mammal remains were devoid of this modification (Table A3). Also present, but not numerously, were calcite precipitations (ca. 0.5%) (n = 20) and roots etching (ca. 0.2%) (n = 10).

Rodent remains were also found in Phase III sediments: namely, a tooth of red squirrel (*Sciurus vulgaris*), two mandibles of yellow-necked field mouse/long-tailed field mouse (*Apodemus flavicollis/sylvaticus*), and three teeth of lagomorphs (hare).

Bird remains in Phase III belong mainly to the domestic chicken; five of them are from immature individuals. Two mallard bones represent young specimen(s). The Eurasian woodcock bone (a coracoid) has a damaged surface on the scapular epiphysis, which likely

happened when a person tore off the bird's shoulder girdle. The falcon bone (a pelvis) has a carnivore tooth puncture. In general, relatively many bird bones in this phase bear traces of gnawing or digestion (Table A5). The digested fragments are too large to be swallowed by a human; domestic dog is presumed responsible.

Phase III is also noted for the largest number of fish bones (n = 245 + 835 scales). Both freshwater and marine (including migratory) fish were discovered. Marine and migratory fish are represented by herring (NISP = 22) and salmonids (NISP = 2). Freshwater fish include the burbot (NISP = 4), perch (NISP = 31), pike (NISP = 21), ide (NISP = 1), roach (NISP = 2), bream (NISP = 2), and undetermined cyprinids (n = 36 + >360 scale fragments); many scales (n = > 430) come from representatives of family Percidae, most probably from the perch (Table 3).

*3.4. Phase IV*

Osteological material related to Phase IV was collected in trenches I, II, IV, and VI and consists of 2425 mammal remains (Table 1). Two taxa of domestic mammals—the pig (NISP = 204) and cattle (NISP = 188)—are best represented and they form ca. 16% of all the mammal osseous material discovered in this phase. Phase IV is important because the largest number of pig (MNI = 9) and goat/sheep (MNI = 4) were recorded in it compared to the other phases.

Similarly to other phases, signs of cut marks on bone surface are not numerous (n = 40) and were found only on bones of two domestic taxa—cattle and goat/sheep (Table A1). The marks are mainly related to dismembering or filleting. Among the remains that could be identified more precisely, chopping marks are much more frequent (n = 160) and appear on bones of domestic species such as cattle, pig, and goat/sheep (Figure 8a). It should also be mentioned that signs of human activity (chopping) are visible on the wild boar maxilla. The total number of burned bones is small (n = 10).

Gnawing marks caused by the dog and signs of digestion were found mostly on the bones of domestic taxa. However, we note that one of the dog bones has gnawing marks made by a carnivore and five bones of hare were digested. Trampling marks were very rare (n = 6) on mammal remains from this phase (Table A2).

Modifications from weathering (n = 17) and plant root activity (n = 16) were noted on the mammal bone surfaces as well. Calcite precipitation was noticed only on the distal surface of pig's long bone (Table A3).

One fragment of the European mole (*Talpa europaea*) was found in a soil sample related to Phase IV.

Bird remains attributed to Phase IV are mostly from domestic chicken and goose (Table 2, Figures 5 and 6). Only singular finds are from mallard duck and house sparrow. Both galliforms and geese include bones from immature specimens (15 bones of the former and two of the latter). It was possible to determine the sex in the case of seven bones of chicken (five bones of female and two of male). Some bones from this phase bear traces of animal activity (gnawing and digesting) (Table A5). One chicken femur has a chopping mark. A cervical vertebra of a middle-sized galliform (probably chicken) has a deep cut mark on its ventral surface, and the cut continues on the lateral side, demonstrating the bird had its throat sliced.

In the sediments of Phase IV, an eel vertebra was found (NISP = 1), as well as fragments of freshwater fish remains which represent pike (NISP = 3), roach (NISP = 1), tench (NISP = 1), dace (NISP = 1), and undetermined cyprinids (NISP = 3). The fish scales are from family Percidae (n = 15) and undetermined fish (n = 15), and the bone fragments (n = 11) of undetermined fish are those mostly from the fins and ribs (Table 3).

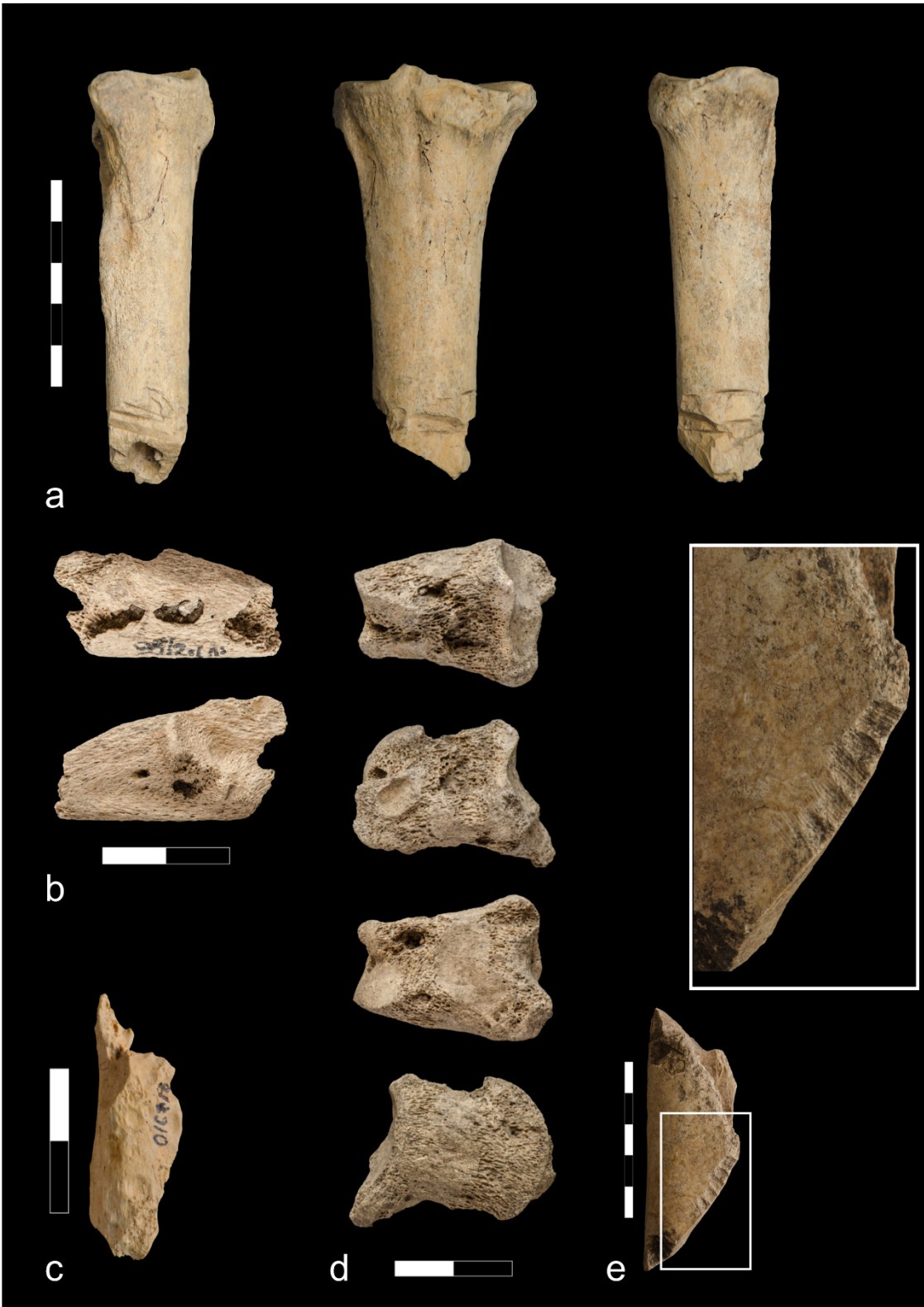

**Figure 8.** Bones from the Ojców castle: (**a**) Cattle metacarpal bone with lateral chopping marks around the shaft (phase IV); (**b**) Digested goat/sheep calcaneus (phase V); (**c**) Digested medium size mammal humerus (phase V); (**d**) Digested cattle's phalange II (phase V); (**e**) Long bone shaft of the large size mammal gnawed by rodents (phase V).

### 3.5. Phase V

The sediments of Phase V were registered in all trenches except trench VII, and in outcrops. Mammal remains from this phase are the most fragmented compared with other phases, and only ca. 19% of the bone fragments could be identified more precisely within the Class Mammalia. As in the older phases, cattle remains show the largest number of individuals (MNI = 8), followed by pig remains (MNI = 4). Hare, wild boar, red deer, and a roe deer represent the wild taxa. However, the presence of a femur of a young rabbit is noteworthy here (Table 1), as discussed below. From the soil sample of Phase V, one bone fragment of the European mole was found.

Despite a high fragmentation of osseous material found from Phase V, the highest number of mammal remains with the signs of human activity were recorded from there (Table A1). The cut marks (n = 81) were seen only on the bones of domestic species. However, most of the marks (n = 67) were on unidentifiable bone fragments. The chopping marks are much more frequent (n = 313): they were recorded nearly twice as often on undeterminable bone fragments (n = 219) than on identified bones (n = 94).

Such signs of human activity were also found on bones of the wild boar and hare. Similar to other phases, burnt bones were very rare; only 16 burnt bone fragments were found, including just two identified pig bones.

Gnawing marks made by dog were most frequently found on bones of domestic mammals (n = 103) (Table A2), undoubtedly given to the dogs after humans had butchered the animals. Some cut marks were recorded on bones of wild mammals—hare, roe deer, and red deer—indicating human butchering of those animals as well. Digested remains, probably eaten by dogs, belong both to the wild and domestic mammals (Figure 8b–d; Table A2). We mention here that gnawing marks made by rodents (Figure 8e) in this phase appeared only on bones of wild mammals. As in other phases, trampling marks are rare (Table A2).

Weathering modifications on bone surfaces are not frequent (n = 12) (Table A3). In addition, only one bone was found to have calcite precipitations. Compared to the other phases, the Phase V materials have the largest number of specimens with traces of root etching (n = 25).

The sediments of Phase V were the richest in bird remains at the site (Table 2). The majority of the bones come from chicken and goose; it is also probable that bones identifiable only as middle-sized galliforms also come from chicken. Thirty-three remains of galliforms (including chicken) are from immature birds; traces on them (two dismembering marks and six cut marks) demonstrate utilization of the chicks for food. Of note, one bone with a cut mark was gnawed, maybe by a human. Moreover, one humerus of a juvenile chicken has its distal part charred while the diaphysis is unaffected by fire; this suggests that the chicken wing was separated from the body and grilled or fried. Additionally, two other bones of unidentified birds are black, which might indicate fire treatment (Table A4), although mineral staining in the soil is also possible.

Galliform bones that might have belonged to adult specimens also have traces of human and animal activity (Tables A4 and A5); for example, one partly digested bone has remnants of coprolite still adhering. At least 24 bones of galliforms belong to females. Two galliform bones are deformed due to pathology.

Goose remains are relatively numerous, but they are mainly small bone fragments or poorly diagnostic parts of the skeleton. Two bones belong to female(s) that died in an egg-laying period, i.e., between March and May. Three goose bones belong to immature specimens, and one of them (an ulna) has a cut mark on its diaphysis. Chop marks were noted on goose axial elements (sternum, coracoid), bones of wings (carpometacarpus and ulna), and a leg bone (a femur). A foot phalange has a pathological modification, suggesting the bird had been reared by humans and not caught in the wild [31].

A bone of the peafowl—a fragment of the right tibiotarsus—is not the first peafowl bone discovered at the site: two more bones were recovered during the earlier excavations [1,32]. The bird occurrence in the medieval Europe is usually associated with upper social class of people [15].

Bones of the common kestrel are the left and right humeri and might have belonged to the same subadult individual. A weathered pelvis came from an unspecified falcon. Two wing bones of the Eurasian jay (humerus, ulna) have small tooth punctures, suggesting the bird was either killed or scavenged by a small carnivore like the cat or marten. The Eurasian jay, the magpie, and the jackdaw are synanthropic birds, but not the woodlark. Woodlark is known to be kept by humans for its voice or used as food during the Middle Ages [30].

Only a few freshwater fish remains are associated with this phase. They include pike (NISP = 5) and some fragments come from the family Cyprinidae (n = 3). In addition, one bone of unidentified fish is recorded.

*3.6. Mixed Layers*

The last chronological group is a combination of several mixed, heterogeneous phases. It consists of phases marked as I–II, I–III, I–IV, I–V, II–III, II–IV, II–V, III–IV, III–V, and IV–V, which, in total, constitute a collection of 4408 remains (Table 1). Mixed phases were identified primarily in trench VII, but also in trenches I, II, IV, VI, and IX.

Since this collection of remains comes from unspecifiable phases, details of bone modifications are not useful for the purpose of this paper. The data about the traces on bones are added in the Tables 3, A1 and A2 and are shown also on Figures 9 and 10.

In the mixed layers, there were also a few rodent remains: two maxillae of the yellow-necked mouse/long-tailed field mouse, a tooth of a bank vole (*Clethrionomys glareolus*), and a single tooth of European pine vole (*Microtus subterraneus*).

Mixed sediments contain remains of birds that are also mostly present in well-defined phases as well. The only exception is the goshawk (Table 2). This bird was a female whose bone was in a stratum attributed to Phase III–IV. A sparrowhawk bone comes from Phase III–IV and, thus, increases the MNI of the species at the site. Noteworthy is one bone fragment of goose from Phase II–III: the fragment is a middle part of an ulna and was either sawed or repeatedly cut around its circuit. Those traces may indicate processing the bone for a tool. Other interesting findings are the bone of a subadult mallard (that increases the probability the ducks were domesticated specimens), four darkened chicken bones (either stained by soil or charred by heat), and a radius of the peafowl. The last bone does not increase the overall MNI of peafowl at the site.

Eighty-eight fragments of fish bones and ca. 94 scales were found in layers related to the mixed phase (see Table 3). Two bones belong to marine fish—one to a herring, and one scuta to sturgeon. The rest were from the freshwater taxa: perch (NISP = 2) and pike (NISP = 11), and one find comes from the carp (NISP = 1). Additionally, scales were found belonging to families Percidae (n = 35) and Cyprinidae (n = 16); the latter was also represented by bone fragments (n = 17). Bone fragments (n = 39) and scales (n = 42) not assigned to any certain species were recorded as unidentified fish.

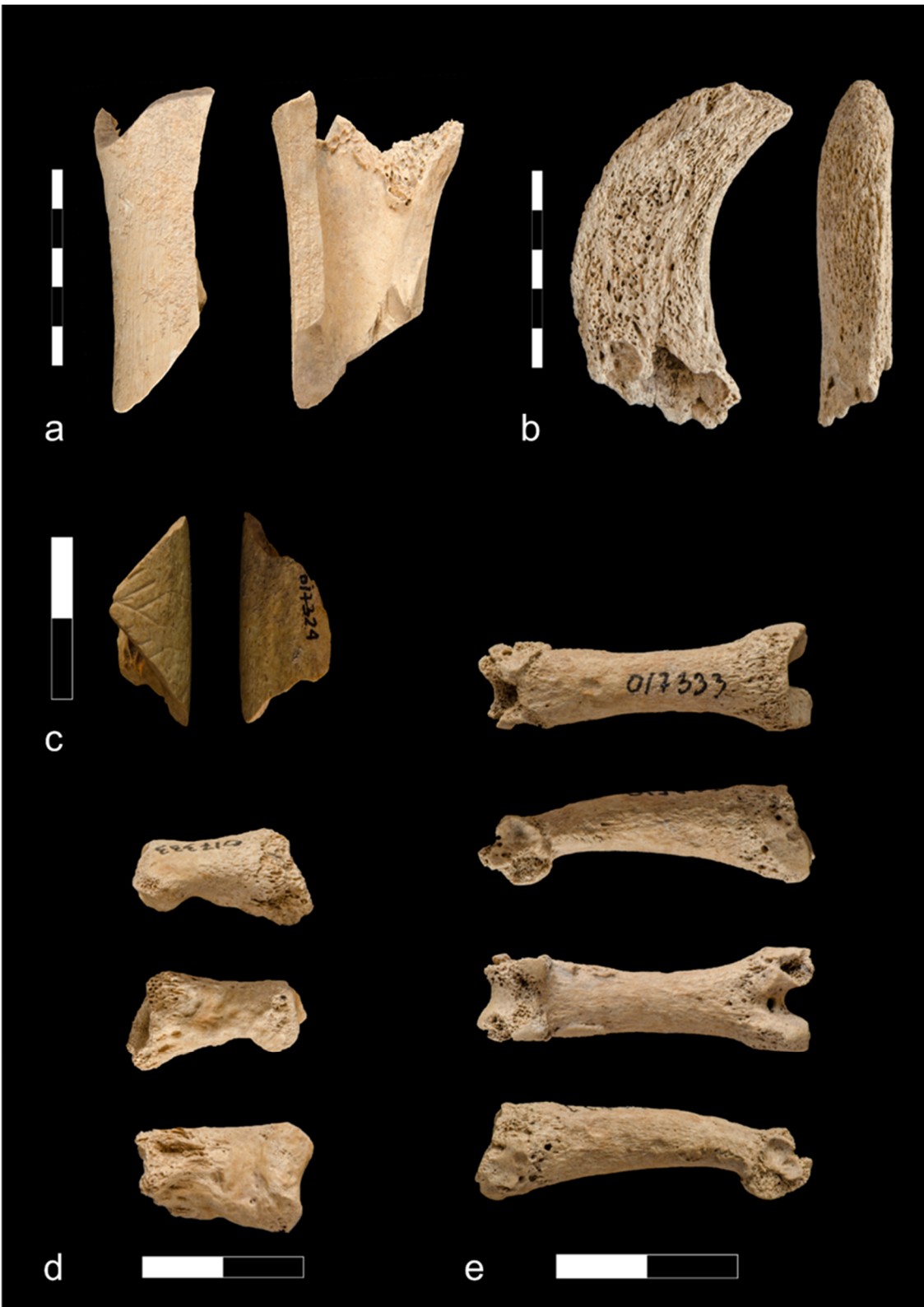

**Figure 9.** Bones from the Ojców castle: (**a**) Plant roots on large size mammal long bone (phase III–IV); (**b**) Sheep poll with sign of grind (phase III–IV); (**c**) Unidentifiable fragment with some ornament (phase II–IV); (**d**) Digested wolf's phalange II (phase II–IV); (**e**) Digested wolf's phalange I (phase II–IV).

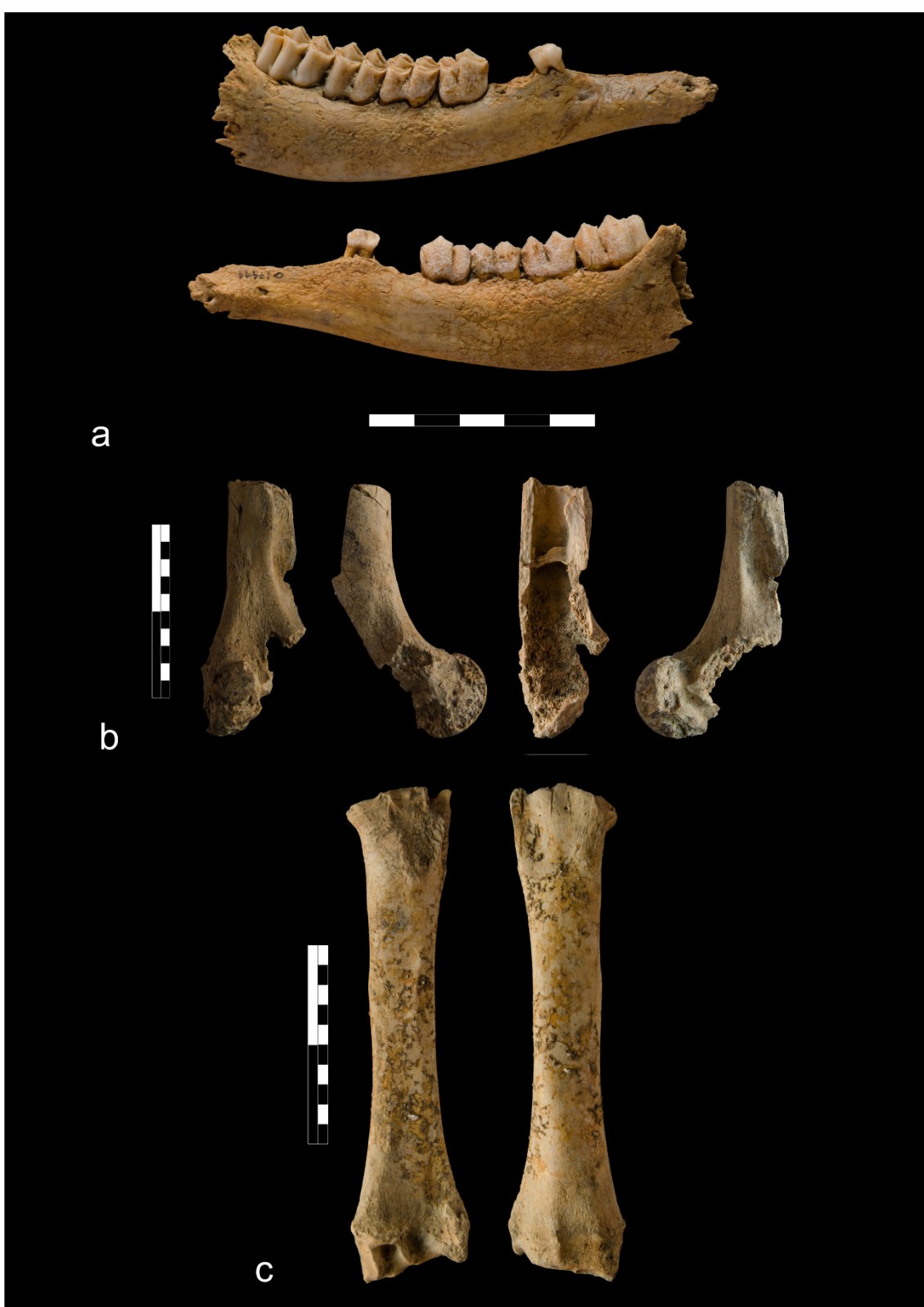

**Figure 10.** Bones from the Ojców castle: (**a**) Plant roots on roe deer mandible (phase II–V); (**b**) Gnawing marks on cattle's femur (phase II–V); (**c**) Calcite efflorescence on cattle's radius (phase II–V).

## 4. Discussion and Conclusions

The animal bones from the Ojców Castle—including those in both the previously published [1] and this new analysis—number over 17,498 remains of mammals, birds, and

fish. From this amount, over 4759 fragments were found from the layers not related to any particular phase. Phase III, which reflects the material from the 15th century, yielded the largest amount of remains, ca. 33% of the total (NISP = 3788), and the largest number of species ca. 30% of the total (MNI = 32). Next, in terms of the number of remains are the sediments of Phase V (second half of 17th century–beginning of 19th century) (ca. 30%; NISP = 3417) and Phase IV (beginning of 16th century–first half of 17th century) (ca. 21%; NISP = 2425). However, the number of individual animals in Phase IV is larger (ca. 27%; MNI = 29) than in Phase V (ca. 22%; MNI = 23). This is a result of the higher fragmentation of the osteological material in the Phase V sediments, reducing our ability to differentiate bones of individuals. The number of mammal bone and bone fragments (NISP = 1490, ca. 13% of total) is also relatively large in Phase I of the site (Early Iron Age and Lusatian culture), and the number of individuals is also large (ca. 17%; MNI = 18).

The results of the previous analysis [1], supplemented by the current study, still indicate the dominance of farm animals throughout all phases. Cattle, pig, and goat/sheep dominate as the main sources of meat. The remains of these animals represent ca. 93% of all the identified bones and teeth. The remaining ca. 3% were identified as horse and dog. Among all domesticated mammals, cattle (MNI = 34 in total) definitely dominate in most of the phases (Table 1, Figures 3 and 4), overshadowing all other taxa. Only in the phase related to the Lusatian culture and in Phase II related to construction of the castle settlement were the numbers of cattle individuals equaled by numbers of pigs. The second-best represented taxon in sediments of particular phases is pig. The third most numerous taxon is goat or sheep, whose minimum number of individuals is more or less the same in all phases except Phase II, where goat/sheep do not appear at all. Certainly, those three domesticated species were primary sources of meat during the whole history of the castle, always dominated by cattle.

The small proportion of wild mammals indicates the minor importance of these species in the diet of the castle residents. Their remains comprise only ca. 4% (NISP = 122) of all identified specimens. Among the wild species, hares (MNI = 6) are the most numerous. Their number minimally increases in the two youngest phases (IV and V). Roe deer (MNI = 3), red deer (MNI = 4), and wild boar (MNI = 4) are present through all phases. Other, smaller wild species such as beaver and red fox are represented by single individuals. Their presence is related to the beginnings of the castle (Phase III). It is worth noting that a bear bone was discovered only in sediments of Phase I (the Lusatian settlement). There are also isolated remains of wolf and elk in the site assemblage, but, unfortunately, it is not possible to state to which particular phase they belong. We highlight that rabbit bones were discovered in sediments of Phase V, possibly related to a tradition popularized by Polish king Augustus II [33] of eating meat of this mammal.

The bird bone assemblage in its overall composition resembles other medieval assemblages in the region (e.g., [4,5,34–39]). The most numerous are birds that were or might have been reared by humans (domestic chicken, goose, duck and pigeon); some of the other less common species might have been domesticated or wild-caught. The fragments identified as middle size galliforms probably also represent domestic chickens. The chicken is a domesticated bird, occurring in nature in South East Asia. The domestic goose (*Anser anser forma domestica*) is a domesticated form of the wild greylag goose (*Anser anser*); the latter occurs naturally in Europe and happens to breed sporadically with its domestic counterpart [40]. Unfortunately, much as the domesticated form is unique in its white plumage, it is hard to identify by a singular bone. The remains of both wild and domestic kinds partially overlap in size and are morphologically similar. Making matters worse, bones of other geese species such as the bean goose and the white-fronted goose also resemble the greylag morphologically and overlap in size [41]. Geese were valued game birds in the Middle Ages [30]. Two other species that were domesticated, the mallard and the rock pigeon, pose similar identification problems to the graylag goose. The domestic duck is usually hard to differentiate from a wild mallard just by the bones, and the bones of the rock pigeon resemble two other pigeon species that live in the wild in the region.

Worth noticing is the presence of the birds of prey, which may suggest falconry was practiced. The goshawk is not a synanthropic bird and, thus, was most likely deposited by a human. Falconry at the site may be further suggested by the presence of the immature common kestrel. Juvenile falcons might have been picked from nests by people and trained for falconry [15]; some might have died during the process. However, it is possible that the common kestrel, which may nest in urban areas, died without human interference. This scenario seems to be quite likely in the case of the house sparrow, the magpie, and the jackdaw, common synanthropic birds. Birds such as the hazel grouse, the capercaillie, and the Eurasian woodcock were valued as game to be hunted during the Middle Ages [30] and have remained such in the present day.

On the basis of the studies of the osteological remains, some differences related to the phases of settlement at the Ojców castle site can be observed. First of all, the Lusatian culture settlement (Phase I) is distinguished from the castle period (Phases II to V). In the case of the Lusatian settlement, there is no apparent deviation in mammal livestock management, which is confirmed by studies at other sites of this type in Lesser Poland [42]. This indicates that the mammal livestock management in this culture was homogeneous and did not undergo strong changes during its duration. However, the high number of chicken bones seems perplexing. Although the bird appeared in central Europe in the seventh–sixth century BCE, it is believed to have become popular later only under the Roman cultural influence there [43].

After analyzing the changes in the species composition during Phases II–V, we cannot clearly state in which phase the settlement of the castle underwent its greatest flowering. The species diversity of mammals increases in Phase III (15th century) and slightly decreases in the subsequent younger phases. This may be directly related to hunting privileges (the so-called hunting regale), which imposed the privilege of hunting mostly large wild game (such as aurochs, European bison, wild boar, bear, deer, roe deer, but also beaver) [30]. In the Phase III material, we are dealing with deer, roe deer, wild boar, beaver, and falcon, which was closely associated with hunting. This indicates that the inhabitants of that time had rights to hunt this game. Additionally, in Phase III, we find the highest number of remains of not only freshwater but also marine fish, attesting to the castle's heyday and economic diversity. The fish subassemblage includes herring, which, until the 15th century, was still a commodity of the transit trade on the north–south route [44]. Single remains of salmonids and eel have also been found in material from this period. The perspective changes somewhat in more recent times, when in the 18th century rabbit, previously undervalued as a type of meat, appeared on upper class tables [33]. The peafowl also appeared, whose presence at court indicates the pre-status of its inhabitants [32]. Also relevant is the occurrence of carp, which, though found in a mixed layer, indicates an import for the upper class, since carp is not naturally distributed in the waters around the castle and must have been brought from somewhere else. This is the case also for herring and sturgeon, which were definitely imported to the site.

Complementing these data with the history of the castle and its inhabitants, it can be concluded that the relatively most prosperous times at the castle date back to the 15th century, and to modern times from the 2nd half of the 17th century onwards. Changes in culinary tastes are particularly noticeable in younger eras. This is mainly due to the increase in the dynamics of the inhabitants' lives, as well as contacts beyond the borders of the kingdom.

Undoubtedly, in the form of a castle hill, we are dealing with a very interesting object from the point of view of archaeology as well as zooarchaeology. Despite the often-changed owners of the castle (in the Middle Ages and modern times), we are dealing with a homogeneous structure forced by the relief of the terrain. In addition, individual trenches, which were located in different places on the hill, give a reason for further research and study on the function of the various areas on the castle hill in more detail. It is hoped that further archaeological excavations will provide new findings in the study of faunal diversity in this site.

**Author Contributions:** Conceptualization, P.W. and J.R.-S.; methodology, P.W., K.W. and L.L.; formal analysis, P.W. and J.R.-S.; investigation, P.W., J.R.-S., K.W., L.L. and A.L.; resources, M.W.; data curation, P.W., J.R.-S., K.W. and L.L.; writing—original draft preparation, P.W., J.R.-S.; writing—review and editing, P.W., J.R.-S., K.W., L.L. and M.W. All authors have read and agreed to the published version of the manuscript.

**Funding:** This research received no external funding in general. Works of L.L. were funded by the Estonian Research Council (grant No. PRG29).

**Acknowledgments:** We are grateful to Gary Haynes for his important comments and his help in English proofreading.

**Conflicts of Interest:** The authors declare no conflict of interest.

## Appendix A

**Table A1.** Traces of human activity on mammal bones.

| Taxon | Phase I Cut Marks | Phase I Chopping | Phase I Burnt | Phase II Cut Marks | Phase II Chopping | Phase II Burnt | Phase III Cut Marks | Phase III Chopping | Phase III Burnt | Phase IV Cut Marks | Phase IV Chopping | Phase IV Burnt | Phase V Cut Marks | Phase V Chopping | Phase V Burnt | Mixed Cut Marks | Mixed Chopping | Mixed Burnt | Total Cut Marks | Total Chopping | Total Burnt |
|---|---|---|---|---|---|---|---|---|---|---|---|---|---|---|---|---|---|---|---|---|---|
| Beaver (*Castor fiber*) | | | | | | | | | | | | | | | | | | | | | |
| Rabbit (*Oryctolagus cuniculus*) | | | | | | | | | | | | | | | | | | 1 | | | 1 |
| Hare (*Lepus europaeus*) | | | | | | | | | | | | | | 1 | | | | | | 1 | |
| Felidae | | | | | | | | | | | | | | | | | | | | | |
| Red fox (*Vulpes vulpes*) | | | | | | | | | | | | | | | | | | | | | |
| Wolf (*Canis lupus*) | | | | | | | | | | | | | | | | | | | | | |
| Dog (*Canis familiaris*) | | 1 | | | | | | | 1 | | | | | | | | | | | 1 | 1 |
| Horse (*Equus caballus*) | | 1 | 1 | | | | | | | | | | | | | | | | | 1 | 1 |
| Roe deer (*Capreolus capreolus*) | | | | | | | | | | | | | | | | | | | | | |
| Red deer (*Cervus elaphus*) | | 1 | | | | | | 3 | | | | | | | | | | | | 4 | |
| Elk (*Alces alces*) | | | | | | | | | | | | | | | | | | | | | |
| Goat/sheep (*Capra hircus*/*Ovis aries*) | 3 | 3 | 1 | | | | 2 | 10 | | 1 | 8 | 3 | 4 | 9 | | 3 | 14 | 4 | 13 | 44 | 8 |
| Cattle (*Bos taurus*) | 6 | 13 | 2 | | 1 | | 9 | 78 | 6 | 5 | 55 | 1 | 5 | 65 | | 12 | 115 | 7 | 37 | 327 | 16 |
| Pig (*Sus domesticus*) | 5 | 7 | 1 | | | 1 | 4 | 27 | 4 | | 12 | 2 | 5 | 18 | 2 | 2 | 34 | 9 | 16 | 98 | 19 |
| Wild boar (*Sus scrofa*) | | | | | | | | | | | 1 | | | 1 | | | | | | 2 | |
| Bear (*Ursus arctos*) | | | | | | | | | | | | | | | | | | | | | |
| **Identifiable total** | **14** | **26** | **5** | | **1** | **1** | **15** | **118** | **11** | **6** | **76** | **6** | **14** | **94** | **2** | **17** | **163** | **21** | **66** | **478** | **46** |
| Small sized mammals | 3 | | 2 | | | | | | | 2 | | | 3 | 1 | | 7 | | 1 | 15 | 1 | 3 |
| Medium sized mammals | 17 | 7 | 10 | 1 | 1 | | 6 | 24 | 3 | 8 | 4 | 2 | 16 | 39 | 3 | 24 | 22 | 16 | 72 | 97 | 34 |
| Large sized mammals | 25 | 30 | 8 | 1 | 6 | | 25 | 112 | 6 | 21 | 63 | 2 | 47 | 161 | 7 | 90 | 135 | 13 | 209 | 507 | 36 |
| Unidentifiable | 3 | 7 | 16 | | | | 6 | 19 | 4 | 3 | 17 | | 1 | 18 | 4 | 14 | 28 | 28 | 27 | 89 | 52 |
| **Unidentifiable total** | **48** | **44** | **36** | **2** | **7** | | **37** | **155** | **13** | **34** | **84** | **4** | **67** | **219** | **14** | **135** | **185** | **58** | **323** | **694** | **125** |
| **TOTAL** | **62** | **70** | **41** | **2** | **8** | **1** | **52** | **273** | **24** | **40** | **160** | **10** | **81** | **313** | **16** | **152** | **348** | **79** | **389** | **1172** | **171** |

**Table A2.** Traces on mammal bones connected to predator and rodent activity.

Column abbreviations: DG = Dog Gnawing, DB = Digested Bones, RG = Rodent Gnawing, TM = Trampling Marks.

| Taxon | Phase I | | | | Phase II | | | | Phase III | | | | Phase IV | | | | Phase V | | | | Mixed Phases | | | | Total | | | |
|---|---|---|---|---|---|---|---|---|---|---|---|---|---|---|---|---|---|---|---|---|---|---|---|---|---|---|---|---|
| | DG | DB | RG | TM | DG | DB | RG | TM | DG | DB | RG | TM | DG | DB | RG | TM | DG | DB | RG | TM | DG | DB | RG | TM | DG | DB | RG | TM |
| Beaver (*Castor fiber*) | | | | | | | | | | | | | | | | | | | | | 1 | | | | 1 | | | |
| Rabbit (*Oryctolagus cuniculus*) | | | | | | | | | | | | | | | | | | | | | | | | | | | | |
| Hare (*Lepus europaeus*) | | | | | | | | | | 2 | | | | 5 | | | 1 | 1 | | | | 2 | | | 1 | 10 | | |
| Felidae | | | | | | | | | | | | | | | | | | | | | | | | | | | | |
| Red fox (*Vulpes vulpes*) | | | | | | | | | | | | | | | | | | | | | | | | | | | | |
| Wolf (*Canis lupus*) | | | | | | | | | | | | | | | | | | | | | | 2 | | | | 2 | | |
| Dog (*Canis familiaris*) | 1 | | | | | | | | 2 | | | 2 | | | | | | | | | 1 | 1 | | | 4 | 1 | | 2 |
| Horse (*Equus caballus*) | | | | | | | | | 3 | | | | 2 | | | | | | | | 1 | | | | 6 | | | |
| Roe deer (*Capreolus capreolus*) | | | | | | | | | 1 | | | | 1 | | | | | | 1 | | 1 | | | | 3 | | 1 | |
| Red deer (*Cervus elaphus*) | | | | | | | | | 2 | | | | 2 | 2 | | | | | | | | | | | 4 | 2 | | |
| Elk (*Alces alces*) | | | | | | | | | | | | | | | | | | | | | | | | | | | | |
| Goat/sheep (*Capra/Ovis*) | 11 | 2 | 2 | 2 | | | | | 34 | 25 | | 1 | 28 | 41 | | | 23 | 20 | 1 | | 42 | 11 | | | 138 | 99 | 3 | 3 |
| Cattle (*Bos taurus*) | 15 | | 2 | | 2 | | | | 134 | 4 | | 10 | 59 | 2 | | 4 | 47 | 3 | | 1 | 103 | 4 | | 4 | 360 | 13 | 2 | 19 |
| Pig (*Sus domesticus*) | 9 | | 1 | | | | | | 81 | 34 | 1 | 4 | 33 | 22 | | 1 | 31 | 8 | | 2 | 59 | 13 | | | 213 | 77 | 2 | 7 |
| Wild boar (*Sus scrofa*) | | | | | | | | | 1 | | | | | | | | | | 1 | | 4 | | | | 5 | | 1 | |
| Bear (*Ursus arctos*) | 1 | | | | | | | | | | | | | | | | | | | | | | | | 1 | | | |
| **Identifiable total** | **37** | **2** | **5** | **2** | **2** | | | | **258** | **65** | **1** | **17** | **121** | **70** | | **5** | **107** | **34** | **3** | **3** | **211** | **33** | | **4** | **736** | **204** | **9** | **32** |
| Small sized mammals | | | | | 1 | | | | | 1 | | | 7 | | | | 2 | | | | 14 | 1 | 1 | 1 | 24 | 2 | 1 | 1 |
| Medium sized mammals | 31 | 1 | 4 | | 4 | | | | 165 | 16 | | | 114 | 24 | | | 151 | 10 | 1 | | 166 | 10 | 1 | 1 | 631 | 61 | 6 | 1 |
| Large sized mammals | 35 | | 11 | 3 | 2 | | | 1 | 205 | 8 | 2 | 4 | 94 | 1 | | 1 | 130 | 1 | 1 | 4 | 224 | 2 | 4 | 6 | 690 | 12 | 18 | 19 |
| Unidentifiable | 35 | 4 | 1 | | 1 | | | | 223 | 79 | | 3 | 79 | 21 | | | 84 | 27 | | 2 | 129 | 39 | 2 | | 551 | 170 | 3 | 5 |
| **Unidentifiable total** | **101** | **5** | **16** | **3** | **8** | | | **1** | **593** | **104** | **2** | **7** | **294** | **46** | | **1** | **367** | **38** | **2** | **7** | **533** | **52** | **8** | **7** | **1896** | **245** | **28** | **26** |
| **Total** | **138** | **7** | **21** | **5** | **10** | | | **1** | **851** | **169** | **3** | **24** | **415** | **116** | | **6** | **474** | **72** | **5** | **10** | **744** | **85** | **8** | **11** | **2632** | **449** | **37** | **57** |

**Table A3.** Traces on mammal bones connected to environmental factors.

| Taxon | Phase I | | | Phase II | | | Phase III | | | Phase IV | | | Phase V | | | Mixed Phases | | | Total | | |
|---|---|---|---|---|---|---|---|---|---|---|---|---|---|---|---|---|---|---|---|---|---|
| | Weathering | Calcite Precipitations | Root Etching | Weathering | Calcite Precipitations | Roots Digesting | Weathering | Calcite Precipitations | Root Digesting | Weathering | Calcite Precipitations | Root Digesting | Weathering | Calcite Precipitations | Root Digesting | Weathering | Calcite Precipitations | Root Digesting | Weathering | Calcite Precipitations | Roots Digesting |
| Beaver (*Castor fiber*) | | | | | | | | | | | | | | | | | | | | | |
| Rabbit (*Oryctolagus cuniculus*) | | | | | | | | | | | | | | | | | | | | | |
| Hare (*Lepus europaeus*) | | | | | | | | | | | | | | | | | | | | | |
| Felidae | | | | | | | | | | | | | | | | | | | | | |
| Red fox (*Vulpes vulpes*) | | | | | | | | | | | | | | | | | | | | | |

**Table A3.** *Cont.*

| Taxon | Phase I | | | Phase II | | | Phase III | | | Phase IV | | | Phase V | | | Mixed Phases | | | Total | | |
|---|---|---|---|---|---|---|---|---|---|---|---|---|---|---|---|---|---|---|---|---|---|
| | Weathering | Calcite Precipitations | Root Etching | Weathering | Calcite Precipitations | Roots Digesting | Weathering | Calcite Precipitations | Root Digesting | Weathering | Calcite Precipitations | Root Digesting | Weathering | Calcite Precipitations | Root Digesting | Weathering | Calcite Precipitations | Root Digesting | Weathering | Calcite Precipitations | Roots Digesting |
| Wolf (*Canis lupus*) | | | | | | | | | | | | | | | | | | | | | |
| Dog (*Canis familiaris*) | | | | | | | | | | | | | | | | | | | | | |
| Horse (*Equus caballus*) | | | | | | | 1 | | | | | | | | | | | 1 | **1** | **1** | **1** |
| Roe deer (*Capreolus capreolus*) | | | | | | | | | | | | | | | | | | 1 | | | **1** |
| Red deer (*Cervus elaphus*) | | | | | | | | | | | | | | | | | | | | | |
| Elk (*Alces alces*) | | | | | | | | | | | | | | | | | | 1 | | | **1** |
| Goat/sheep (*Capra hircus*/*Ovis aries*) | | | | | | | 1 | 2 | | | | 2 | | | 1 | 1 | | 1 | **2** | **2** | **4** |
| Cattle (*Bos taurus*) | 1 | | 1 | | 1 | | 32 | 5 | 1 | 11 | | 7 | 6 | | 2 | 27 | 9 | 11 | **77** | **15** | **21** |
| Pig (*Sus domesticus*) | | 2 | | | | | 2 | 4 | 2 | 2 | 1 | | 1 | 1 | 7 | 1 | 3 | 1 | **6** | **9** | **12** |
| Wild boar (*Sus scrofa*) | | | | | | | | | | | | | 1 | | | | | | **1** | | |
| Bear (*Ursus arctos*) | | | | | | | | | | | | | | | | | | | | | |
| **Identifiable total** | **1** | **2** | **1** | | | | **36** | **11** | **3** | **13** | **1** | **9** | **8** | **1** | **10** | **29** | **12** | **16** | **87** | **26** | **40** |
| Small sized mammals | | | | | | | | | | | | | | | | | | 2 | | | **2** |
| Medium sized mammals | 1 | 1 | 1 | | | | 1 | 4 | 2 | | | 1 | | | 3 | | 4 | 7 | **2** | **9** | **14** |
| Large sized mammals | 3 | 4 | | | | | 7 | 4 | 5 | 3 | | 6 | 3 | | 11 | 5 | 11 | 11 | **21** | **15** | **37** |
| Unidentifiable | | | | | 4 | | 1 | 1 | | 1 | | | 1 | | 1 | | 1 | 7 | **3** | **6** | **8** |
| **Unidentifiable total** | **4** | **5** | **5** | | **4** | | **9** | **9** | **7** | **4** | | **7** | **4** | | **15** | **5** | **16** | **27** | **26** | **30** | **61** |
| **Total** | **5** | **7** | **6** | | | | **45** | **20** | **10** | **17** | **1** | **16** | **12** | **1** | **25** | **34** | **28** | **43** | **113** | **56** | **101** |

**Table A4.** Traces of human activity on bird bones.

| Taxon | Phase I | | | Phase II | | | Phase III | | | Phase IV | | | Phase V | | | Mixed Phases | | | Total | | |
|---|---|---|---|---|---|---|---|---|---|---|---|---|---|---|---|---|---|---|---|---|---|
| | Dismembering | Cut Marks | Burnt | Dismembering | Cut Marks | Burnt | Dismembering | Cut Marks | Burnt | Dismembering | Cut Marks | Burnt | Dismembering | Cut Marks | Burnt | Dismembering | Cut Marks | Burnt | Dismembering | Cut Marks | Burnt |
| Goose (*Anser* sp.) | 2 | 1 | | 1 | | | | 1 | | | | | 10 | 5 | | 6 | 1 | | **19** | **8** | |
| Mallard (*Anas platyrhynchos*) | | | | | | | | | | | | | 1 | | | 1 | | | **2** | | |
| Anseriformes (duck/goose) | | | | | | | | | | | | | 1 | | | | | | **1** | | |
| Domestic chicken (*Gallus domesticus*) | 10 | 5 | | | | | 2 | | | 2 | 3 | | 13 | 7 | 1 | 11 | 5 | ?4 | **38** | **20** | **1 + ?4** |
| cf. *Gallus domesticus* | | | | | | | | | | | 1 | | 1 | 2 | | 1 | 2 | | **2** | **5** | |
| Galliformes (middle size) | | | | | | | | | | | 1 | | | 6 | | 1 | | | **1** | **7** | |
| Rock pigeon (*Columba livia*) | 1 | 1 | | | | | | | | | | | | | | | | | **1** | **1** | |

**Table A4.** *Cont.*

| Taxon | Chronology by Phases | | | | | | | | | | | | | | | | | | | | | Total | | |
|---|---|---|---|---|---|---|---|---|---|---|---|---|---|---|---|---|---|---|---|---|---|---|---|---|
| | Phase I | | | Phase II | | | Phase III | | | Phase IV | | | Phase V | | | Mixed Phases | | | | | | | | |
| | Dismembering | Cut Marks | Burnt | Dismembering | Cut Marks | Burnt | Dismembering | Cut Marks | Burnt | Dismembering | Cut Marks | Burnt | Dismembering | Cut Marks | Burnt | Dismembering | Cut Marks | Burnt | Dismembering | Cut Marks | Burnt |
| Unidentified bird (Aves indet.) | | | | | | | | | | | | | | 1 | ?2 | | 1 | | | 2 | ?2 |
| **Total** | 13 | 7 | 0 | 1 | 0 | 0 | 2 | 1 | 0 | 2 | 5 | 0 | 26 | 21 | 1 + ?2 | 20 | 9 | ?4 | 64 | 43 | 1 + ?6 |

**Table A5.** Traces on bird bones connected to predator activity.

| Taxon | Chronology by phases | | | | | | | | | | | | Total | |
|---|---|---|---|---|---|---|---|---|---|---|---|---|---|---|
| | Phase I | | Phase II | | Phase III | | Phase IV | | Phase V | | Mixed Phases | | | |
| | Gnawed | Digested | Gnawed | Digested | Gnawed | Digested | Gnawed | Digested | Gnawed | Digested | Gnawed | Digested | Gnawed | Digested |
| Goose (*Anser* sp.) | | | | | | 3 | | 1 | 8 | 1 | 4 | 1 | **12** | **6** |
| cf. *Anser* sp. | | | | | | | | | 1 | | | | **1** | |
| Mallard (*Anas platyrhynchos*) | | | | | | | | | | | | 1 | | **1** |
| cf. *Anas platyrhynchos* | | | | | | | | | | | | 1 | | **1** |
| Peafowl (*Pavo cristatus*) | | | | | | | | | | | 2 | | **2** | |
| Domestic chicken (*Gallus domesticus*) | | | | | 3 | 3 | 8 | 2 | 8 | 5 | 9 | 2 | **28** | **12** |
| cf. *Gallus domesticus* | | | | | | | | | | 1 | 1 | | **1** | **1** |
| Capercaillie (*Tetrao urogallus*) | | | | | | 1 | | | | | | | | **1** |
| Galliformes (middle size) | | | | | | | | | 2 | 1 | 1 | 1 | **3** | **2** |
| a Falcon (*Falco* sp.) | | | | | 1 | | | | | | | | **1** | |
| Jackdaw (*Corvus monedula*) | | | | | | | | | 2 | | | | **2** | |
| Unidentified bird (Aves indet.) | | | | | 2 | 4 | 1 | 5 | 2 | | 3 | 1 | **8** | **10** |
| Aves indet. (big size) | | | | | | | | | | | 1 | | **1** | |
| cf. Aves indet. | | | | | | | | | | | 1 | | **1** | |
| **Total** | 0 | 0 | 0 | 0 | 6 | 11 | 9 | 8 | 23 | 8 | 22 | 7 | **60** | **34** |

**Table A6.** Traces on bird bones connected to environmental factors.

| Taxon | Chronology by Phases | | | | | | | | | | | | | | | | | | | | | | | | Total | | | |
|---|---|---|---|---|---|---|---|---|---|---|---|---|---|---|---|---|---|---|---|---|---|---|---|---|---|---|---|---|
| | Phase I | | | | Phase II | | | | Phase III | | | | Phase IV | | | | Phase V | | | | Mixed Phases | | | | | | | |
| | Root Etching | Weathering | Trampling | Rodent Gnawing | Root Etching | Weathering | Trampling | Rodent Gnawing | Root Etching | Weathering | Trampling | Rodent Gnawing | Root Etching | Weathering | Trampling | Rodent Gnawing | Root Etching | Weathering | Trampling | Rodent Gnawing | Root Etching | Weathering | Trampling | Rodent Gnawing | Root Etching | Weathering | Trampling | Rodent Gnawing |
| Goose (*Anser* sp.) | | | | | | | | | | | | | | | | | 17 | 1 | 4 | 1 | 4 | | | | **21** | **1** | **4** | **1** |
| cf. *Anser* sp. | | | | | | | | | | | | | | | | | 1 | | | | 1 | | | | **2** | | | |
| Mallard (*Anas platyrhynchos*) | | | | | | | | | | | | | | | | | 1 | | | | 4 | | | 1 | **5** | | | **1** |
| Peafowl (*Pavo cristatus*) | | | | | | | | | | | | | | | | | 1 | | | | | 1 | | | **1** | **1** | | |

**Table A6.** *Cont.*

| Taxon | Phase I | | | | Phase II | | | | Phase III | | | | Phase IV | | | | Phase V | | | | Mixed Phases | | | | Total | | | |
|---|---|---|---|---|---|---|---|---|---|---|---|---|---|---|---|---|---|---|---|---|---|---|---|---|---|---|---|---|
| | Root Etching | Weathering | Trampling | Rodent Gnawing | Root Etching | Weathering | Trampling | Rodent Gnawing | Root Etching | Weathering | Trampling | Rodent Gnawing | Root Etching | Weathering | Trampling | Rodent Gnawing | Root Etching | Weathering | Trampling | Rodent Gnawing | Root Etching | Weathering | Trampling | Rodent Gnawing | Root Etching | Weathering | Trampling | Rodent Gnawing |
| Domestic chicken (*Gallus domesticus*) | | | | 1 | | | | | 6 | | 3 | | 6 | 1 | | 4 | 32 | 2 | | 9 | 22 | 3 | 2 | 3 | **66** | **6** | **5** | **17** |
| cf. *Gallus domesticus* | | | | | | | | | | | | | 2 | | | | | | | | 2 | | | 1 | **4** | | | **1** |
| Galliformes (middle size) | | | | | | | | | | 1 | | | 1 | | | | 8 | | | 1 | 1 | | | 1 | **10** | **1** | | **2** |
| a Pigeon (*Columba* sp.) | | | | | | | | | | | | | | | | | 1 | | | | | | | | **1** | | | |
| Common kestrel (*Falco tinnunculus*) | | | | | | | | | | | | | | | | | 2 | | | | 1 | | | | **3** | | | |
| a Falcon (*Falco* sp.) | | | | | | | | | | | | | | | | | 1 | 1 | | | | | | | **1** | **1** | | |
| Eurasian jay (*Garrulus glandarius*) | | 1 | | | | | | | | | | | | | | | | | | | | | | | | **1** | | |
| Magpie (*Pica pica*) | | | | | | | | | | | | | | | | | | | | | | | | 1 | | | | **1** |
| Jackdaw (*Corvus monedula*) | | | | | | | | | | | | | | | | | 2 | | | | | | | | **2** | | | |
| a Corvid (Corvidae indet.) | 1 | | | | | | | | | | | | | | | | | | | | | | | | **1** | | | |
| Unidentified bird (Aves indet.) | 1 | | | | | | | | 2 | | 1 | | 1 | | | | 3 | | 1 | | 2 | 1 | | | **9** | **1** | **2** | |
| Aves indet. (big size) | | | | | | | | | | | | | | | | | 1 | | | | | | | | **1** | | | |
| cf. Aves indet. | | | | | | | | | 1 | | 1 | | | | | | | | | | 2 | | | | **3** | | **1** | |
| **Total** | **2** | **1** | **0** | **1** | **0** | **0** | **0** | **0** | **9** | **1** | **5** | **0** | **10** | **1** | **0** | **4** | **70** | **4** | **5** | **12** | **39** | **5** | **2** | **6** | **130** | **12** | **12** | **23** |

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
