# Peer review of "Zooarchaeological Evidence from Medieval Ojców Castle, Lesser Poland"

_heritage, doi:10.3390/heritage6010013_

Round 1

Reviewer 1 Report

First of all, my compliments for writing this article, which includes a detailed presentation of an important faunal assemblage, and would benefit the work of future research.

The article presents a standard zooarchaeological report on faunal assemblage. The authors compile the data of the current research with those from earlier excavations, which is a good point. They present this work as comprehensive. The time span of the occupation of the site, ranging from the Iron Age till the 19th century is quite extended. Although all together the whole faunal assemblage is quite big, the sub-samples for each phase resulted in a quite limited amount of remains, which hamper significative conclusions.

The article is very informative and extremely (sometimes too extensively) descriptive in the presentation of the data and methods. This part overextends the one on conclusions. As stated in the introduction the paper aimed to be a comprehensive overview per phases with great expectations in the interpretations and insights of the lives of the ‘inhabitants’ of the castle, and of previous occupation, also making references to written sources. In the ‘Conclusion’, however, the ‘everyday life’ of the ‘inhabitants’ does not really emerge. The inhabitants are not well-defined. Is it possible to define who were the inhabitants of the castle or of the settlement? Nobility? but also servants or others… Is it possible to define the archaeological context (refuse pits or another feature) from where the bones are coming? Is the bone material considered as all food refuse? In its whole discarded in the courtyard?  The part on the ‘Conclusion’ would benefit from continuous reflection and reference to what is written as expectations in the ‘Introduction’.  The text would also benefit from more integration of the correlation with the written sources in the text itself. In one case it has been made, (see lines 556-558 about the rabbit in phase V).

It is not clear whether the material is mainly hand-collected or not. Lines 97-98: there is a reference to the taken of soil samples for wet-sieving in order to retrieve microfauna. Are the data presented, especially those of birds and fish, all hand collected or part of these data/remains are coming from sieved samples?

The data/results are presented in clear tables and widely described in the texts. For a better and easier comprehension of the data for the reader, bar histograms would be of great help for visualizing and for a better comprehension of the changes or continuities in the pattern of species present.

The description of the species per phase is mainly based on referring percentages and evidence of human traces. Are no information on the age at the slaughter of the animals? It would be interesting to see whether there is a selective preference for meat between, for example, the pre- and during castle occupation phase.

Some remarks in the text:

504 comprehensive overview: repetition of the aims of the paper, actually already given in previous parts.

535 greatly increases: actually vague; how much does it increase?

540: MNI 34 on the whole sample /phases. Personally, I think that the use of a MNI on a large scale of excavation and of time, without a defined context, is methodologically not a strong valid figure for the comprehension of the contribution of a species to the diet

622 too general conclusion. Why is it so ‘interesting’? And why so ‘undoubtedly’? I would make these statements more supported by explanations of the importance and unicity of the sample as the authors consider to be.

Reviewer 2 Report

The paper by Religa-Sobczyk and colleagues presents the results of the zooarchaeological analysis of the complete assemblage from Medieval Ojców Castle, located in Lesser Poland. The aim is to add information on the diet of the castle inhabitants from the time of establishment until the final residents. This is further complemented by adding zooarchaeological data on older phases of occupation (early Iron Age Lusatian culture). Previously published data are included in this paper, thus also being a synthesis of available information.

The “Introduction” focus on the Ojców Castle, giving a thorough characterisation of the historical data. The “Materials and Methods” briefly presents the different excavation areas, and the methodologies adopted for the zooarchaeological and taphonomical analysis. A third point called “Chronology of the sediments” clarifies the phases discussed in the manuscript. The needed background for the understanding of the site under analysis is given considering the introductory, materials, and chronology sections. The “Results” section clearly and concisely presents the data obtained in general and by different occupation phases. “Discussion and conclusions” further discuss some aspects, summarizing the results and interpretations coherently. We must emphasize that further data, concerning traces of human, predator, and rodent activity, as well as traces connected to environmental factors, are presented in six different tables in the Appendix section.

This is a scientifically sound manuscript, with a very complete methodological framework. Such an in-depth taphonomical analysis is unusual for assemblages of these periods and this is fantastic. The discussion and conclusions are strong and consistent with the pieces of evidence presented. The figures, tables and captions are informative and well-prepared. The bibliography is well-elaborated and includes updated references that deal with the scope of the paper. Overall, this article fits the journal’s scope and standards.

I want to address two recommendations to the authors:

·    I suggest that a small introductory paragraph is added at the beginning of the “Introduction” section quickly mentioning other “roughly contemporaneous” sites/faunal studies in the region. Although the authors present a generic comparison of the results obtained with other zooarchaeological assemblages (lines 558-590), more information on that is needed for the international reader to better understand the broader topics under study;

·    I suggest that the section “Chronology of the sediments” is included in the more general “Materials and methods” section, in the part where the site’s stratigraphy and excavations are presented. This will help on leading the reader through the site’s archaeological characterization before passing to the more “zooarchaeological part” of the paper.

A few minor corrections are needed:

·    Table 1. Check the italicization of “Felidae”

·    Table 2. Check the capitalization of “pigeon”, “falcon”, and “corvid”

·    Line 226. Should it be “the bird bone assemblage”?

·    Line 234. Check the italicization of “sp.”

·    Lines 334, 496, and 532. Check the italicization of “ca.”

·    Line 441. Check “aso”

·    Figure 6. Check the wolf’s phalanges relation between the figure and the caption

·    Line 544. Do you mean “whose minimum number of individuals”?

·    Line 644. Should be “.” not “,”

·    Line 656. Delete “]”

These go beyond the scope of the present article, but, If I may, I want to address some further comments and suggestions to the authors that they may want to consider for future works.

The authors present modifications made by carnivores and infer that they are the result of domestic dogs’ actions (e.g., lines 138-141; 258-260). This is the most parsimonious explanation and probably applies to the assemblage, but since swine are numerically important in this collection, I wonder if, for example, the typical score marks made by swine are completely absent. Swine can produce important modifications on bones during feeding but are commonly under-considered (e.g., [1][2]).

The authors state that “gnawing traces by other animals could not always be distinguished among possible agents affecting smaller bones (namely dogs, cats, and humans)” (lines 161-163); falconry is also suggested (lines 575-585); and small carnivore action is mentioned (e.g., 465-468). In fact, besides larger animals, rabbit/hare digested bones were also identified, as well as other small animals (birds). Although it is not easy to talk about human action BSM (bone surface modifications) in these smaller animals (e.g., [3]), by discarding a large influence of other predators we can emphasize this possibility. It seems that the assemblage presents BSM from different agents, even if their proportion/influence is unequal. The presence of possible human tooth marks (lines 444-445) is noteworthy and could be even more common in the assemblage. You might already know their existence but interesting data from several colleagues could help further understanding human chewing in remains of animals the size of caprine (e.g., [4][5]).

I believe there is space for analysis of tooth marks evidence (e.g., size) in this assemblage and want to encourage the authors to look at this in future works and to go even deeper regarding the characterisation of consumption patterns ofOjców Castle fauna. Considering the results from your manuscript it would undoubtedly be interesting to further deepen our understanding of such an interesting assemblage.

Finally, by presenting data from such a large collection I understand why other possible aspects are not presented or further discussed. I believe that is the case of kill-off and butchering patterns, two aspects that could eventually be interesting when comparing the different phases of occupation of Ojców Castle. I think that these deserve further pondering and eventually publication, especially if differences are seen between occupation phases.

It was a complete pleasure to read your manuscript. I want to congratulate you on your work and I am looking forward to new developments. Gratulacje!

[1] https://doi.org/10.1179/jfa.1988.15.4.473

[2] https://doi.org/10.1002/oa.987

[3] https://doi.org/10.1007/s12520-022-01662-8

[4] https://doi.org/10.1016/j.jhevol.2010.08.003

[5] https://doi.org/10.1016/j.jas.2012.08.002
